# Permutree Process

## Abstract

This paper presents a Bayesian nonparametric (BNP) method based on an innova-
tive mathematical concept of the *permutree*, which has recently been introduced in
the field of combinatorics. Conventionally, combinatorial structures such as permu-
tations, trees, partitions and binary sequences have frequently appeared as building
blocks of BNP models, and these models have been independently researched and
developed. However, in practical situations, there are many complicated problems
that require master craftsmanship to combine these individual models into a single
giant model. Therefore, a framework for modelling such complex issues in a unified
manner has continued to be demanded. With this motivation, this paper focuses for
the first time in the context of machine learning on a tool called the permutree. It
encompasses permutations, trees, partitions, and binary sequences as its special
cases, while also allowing for interpolations between them. We exploit the fact that
permutrees have a one-to-one correspondence with special permutations to propose
a stochastic process on permutrees, and further propose a data modeling strategy.
As a significant application, we demonstrate the potential for phylogenetic analysis,
which involve coalescence, recombination, multiple ancestors, and mutation.

## 1 Introduction

**Various combinatorial structures** - *Permutations*, *trees*, *partitions*, and *binary sequences* have been
frequently utilized in Bayesian modeling, and conventionally, various models have been studied
separately for each subject. *Permutations* have been used in a wide range of applications such
as Bayesian ranking [101, 63, 110, 73], matrix reordering [70, 81, 99], and the traveling salesman
problem [102, 17, 105]. Various random permutation models, such as the Mallows model [54, 12, 16],
the permuton models [37, 7, 51, 6] and the modified Chinese restaurant process [57], have been
employed in Bayesian modeling. *Trees* are typically used for hierarchical clustering [22, 21] and
multiple resolution regression [47, 25, 20]. In the Bayesian literature, the Dirichlet diffusion tree [62,
45], the Mondrian process [88, 87] and the Pólya tree [56, 48, 27, 15] are particularly popular models.
*Partitions and binary sequences* are fundamental tools in machine learning, with numerous examples
of their usage in clustering, factor analysis, feature selection, and more. For the modeling of partitions
and binary sequences, the Dirichlet process mixture model [23, 79, 100, 59], the Pitman-Yor process
mixture model [76], the Chinese restaurant process [74], and the stick-breaking process [89] for
random partitions, and the beta-Bernoulli process and the Indian buffet process [29, 95, 93] for
random binary sequences have frequently been employed.

**Combination of different combinatorial structures** - In real-world applications of machine learning,
it is often a useful strategy to combine several different combinatorial structures to model data, rather
than using only one combinatorial structure. For example, the combination of partitioning and factor
models is particularly popular, including the infinite factorial hidden Markov models [24, 97], the
subset infinite relational model [39], the infinite latent factor model with the infinite mixture model
[111, 112] and the kernel beta process [83]. As another example, the combination of tree structures

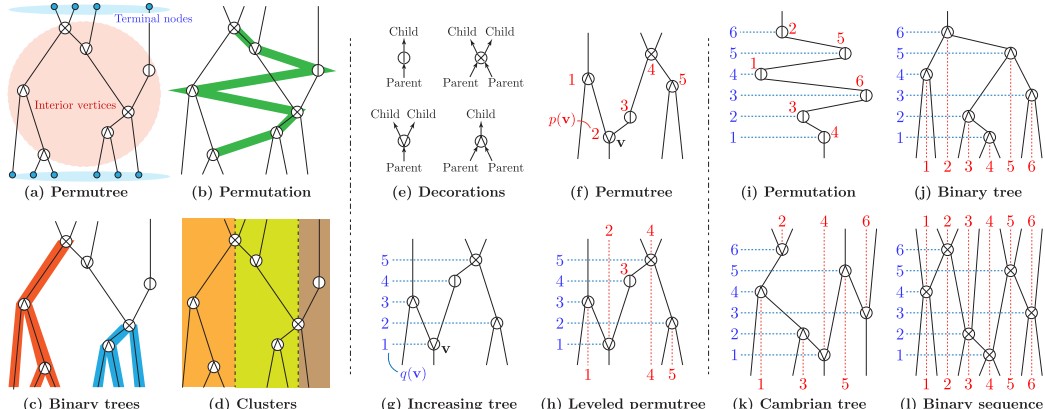

Figure 1: Overview of new combinatorial structures invented in [75]. **Left**: A permutree that is a combinatorial object that includes the concepts of permutations, binary trees, clusters, factors, etc., but can also interpolate between them. The permutree is, as defined, a "directed" tree, but for visibility, the direction of the edges from Parent to Child is omitted in the diagrams. **Middle**: Variant concepts required to represent stochastic processes on permutrees indirectly through *decorated permutations* in Section 3. **Right**: Special cases of permutrees. Remark 2.1 provides details on each interpretation.

and partitioning have also been actively studied, including the hierarchical Dirichlet process [94], the nested Dirichlet process [84, 64], their hybrid models [2, 71, 50], the infinite context-free grammar [49] and the tree-structured stick-breaking process [1, 65]. Furthermore, permutations are occasionally employed in conjunction with clustering to analyze relational data [61]. As we have discussed so far, this kind of strategy of combining multiple models into a single model is one promising direction for research and development. However, advancing research in this direction necessitates the evaluation of an enormous number of models in a combinatorial fashion, which becomes infeasible due to the exponentially increasing number of potential combinations. Consequently, we are striving to initiate a paradigm shift towards exploring an entirely new approach capable of unifying these models.

**Key insight** - In our pursuit of creating a unified model capable of encompassing permutations, trees, partitions, and binary sequences, we are incorporating the concept of *permutrees* [75], which has recently emerged in the field of combinatorics, into the realm of Bayesian nonparametric (BNP) machine learning. Permutrees not only serve as a framework that includes permutations, trees, partitions, and binary sequences as distinct cases but also exhibit intriguing properties of interpolation between them. Figure 1 (a)-(d) provides a concise visual representation of the key characteristics.

**Our contributions** - The main contribution of this paper is to produce, by using the concept of permutrees, a stochastic process that can represent combinatorial structures such as permutations, binary trees, partitions and binary sequences in a unified manner for the first time. Section 3 exploits the one-to-one correspondence between permutations and certain permutations using a two-dimensional marked point process to construct this process, which we call a *permutree process*. Section 4 derives a data modelling strategy using this stochastic process by analogy with the stick-breaking process that is frequently used in BNP machine learning. Section 5 demonstrates the application of phylogenetic analysis of DNA sequence data dealing with multiple biological events such as coalescence, recombination, mutation and multiple ancestry in a unified manner.

## 2 Preliminaries: Permutree and related objects

**Permutree** [75] - A *permutree* is a new mathematical tool invented recently in the field of combinatorics, which not only represent permutations, trees, partitions, and binary sequences as special cases, but can also interpolate between them [75]. Let us begin with the definition of a permutree. We consider a directed tree $\mathbf{T}$ with a vertex set $\mathbf{V}$ of $n$ ($n \in \mathbb{N}$) vertices of degree at least 2, and a set of terminal nodes of degree 1 (See also Figure 1 (a)). For technical reasons (discussed immediately below), we dare to pay particular and explicit attention here to the set $\mathbf{V}$ of the "interior vertices" (i.e., vertices of degree at least 2) other than the terminal nodes. Each vertex $\mathbf{v} \in \mathbf{V}$ is assigned a natural number $p(\mathbf{v})$ as a label, using the bijective vertex labeling (one-to-one correspondence) $p : \mathbf{V} \to [n] := \{1, 2, \ldots, n\}$ based on the following *permutree requirements* (Definition 1 in [75]):

(C1) Each vertex $\mathbf{v} \in \mathbf{V}$ has one or two parents, and one or two children.

(C2)  If a vertex $\mathbf{v}$ has a left parent (or child), then all labels in the subtree of the left ancestor (or descendant) of $\mathbf{v}$ are smaller than $p(\mathbf{v})$. If $\mathbf{v}$ has a right parent (or child), then all labels in the subtree of the right ancestor (or descendant) of $\mathbf{v}$ are greater than $p(\mathbf{v})$.

A directed tree $\mathbf{T}$ that satisfies the above requirements can be expressed more intuitively and clearly by introducing the notion of *decorations* to the vertices $\mathbf{V}$. See also Figure 1 (e). We introduce the $n$-tuple decorations $\delta(\mathbf{T}) := (\delta(\mathbf{T})_1, \ldots, \delta(\mathbf{T})_n) \in \{①, ⊗, ⊘, ⊙\}^n$, defined as follows: (i) $\delta(\mathbf{T})_{p(\mathbf{v})} = ①$ if $\mathbf{v}$ has one parent and one child, (ii) $\delta(\mathbf{T})_{p(\mathbf{v})} = ⊗$ if $\mathbf{v}$ has two parents and two children, (iii) $\delta(\mathbf{T})_{p(\mathbf{v})} = ⊘$ if $\mathbf{v}$ has one parent (lower in Figure 1 (e)) and two children (upper), and (iv) $\delta(\mathbf{T})_{p(\mathbf{v})} = ⊙$ if $\mathbf{v}$ has two parents (lower) and one child (upper). The symbolic feature of permutrees can represent various combination objects in a unified manner as follows:

**Remark 2.1.** *(See Example 4 in [75].)* **Permutation** *- Permutrees with decoration $①^n$ have a one-to-one correspondence with permutations of $[n]$. For example, by reading the* horizontal labels *in the order of the natural number of* vertical labels*, Figure 1 (i) represents a permutation* **436152**. **Binary tree** *- Permutrees with decoration $⊙^n$ have a one-to-one correspondence with rooted planar binary trees on $n$ vertices. See Figure 1 (j) for an example.* **Cambrian tree** *- Permutrees with decoration $\{⊘, ⊙\}^n$ are exactly the Cambrian trees proposed in [82, 13]. See Figure 1 (k) for an example.* **Binary sequence** *- Permutrees with decoration $⊗^n$ have a one-to-one correspondence with binary sequences with length $n - 1$. The $i$th element of the binary sequence is determined according to the following procedure: for any $i \in [n - 1]$, there exists $p(\mathbf{v}) = i$ and $p(\mathbf{w}) = i + 1$, and if $\mathbf{v}$ is the parent of $\mathbf{w}$, output $1$, otherwise output $0$. See Figure 1 (l) for* **10010** *as an example.*

Now that we have summarized the important property of permutrees, we will describe the findings necessary to construct a stochastic process on a permutree, which is the main focus of this paper. As a motivation for describing the following findings, imagine actually drawing an instance of permutree on a hand-drawn blackboard. At this point, we notice that the *horizontal* positional relationship of vertices $\mathbf{V}$ is explicitly given by the natural number label $p(\cdot \in \mathbf{V})$, however, the *vertical* positional relationship is still ambiguous (In Figure 1, (f) is identical to (g) in terms of the permutree, but distinct in terms of the increasing tree). Hence, in order to construct a stochastic process on permutrees in a concise and clear manner, a mechanism to control the vertical positioning of the vertices of the permutrees is required. With this motivation in mind, we introduce two useful notions, an *increasing tree* (Figure 1 (g)) and a *leveled permutree* (Figure 1 (h)).

**Leveled permutree** - To define the leveled permutree, we start by introducing an additional notion of an *increasing tree*. We consider a directed tree $\mathbf{T}$ with vertex set $\mathbf{V}$. Each vertex $\mathbf{v} \in \mathbf{V}$ is assigned a natural number label $q(\mathbf{v})$, using the bijective vertex labeling (one-to-one correspondence) $q : \mathbf{V} \to [n]$ such that, if $\mathbf{v} \in \mathbf{V}$ is the parent of $\mathbf{w} \in \mathbf{V}$, then $q(\mathbf{v}) < q(\mathbf{w})$ is satisfied. Intuitively, the function $q$ serves to label the vertices $\mathbf{V}$ from $1$ to $n$ vertically from bottom to top (Figure 1 (g)). Then, a *leveled permutree* is a directed tree $\mathbf{T}$ with a vertex set $\mathbf{V}$ endowed with two bijective vertex labelings $p, q : \mathbf{V} \to [n]$ which respectively define a permutree and an increasing tree. By using two types of labels $p$ and $q$, the horizontal and vertical arrangement of the vertices $\mathbf{V}$ can be explicitly specified, as shown in Figure 1 (h). The leveled permutree is a useful tool when considering the generative model of the permutree in Section 3, because its specification is clear.

The notion of a leveled permutree so far has improved the prospects for dealing with permutrees. However, leveled permutrees are still combinatorial and geometric, and are not yet easy to handle computationally (in terms of Bayesian modeling, which is the main objective of this paper). Finally, we would like to wrap up this section by revealing one of the most important aspects of leveled permutrees: their relationship to *decorated permutations*.

**Decorated permutation** - For the description of decorated permutations, the notion of a *permutation table* should be prepared first. A permutation table is a geometrical representation of a permutation $\sigma$ with $n$ length by the $(n \times n)$-table, with rows labeled by positions from bottom to top and columns labeled by values from left to right, and with a dot at column $i$ and row $\sigma(i)$ for all $i \in [n]$ [9]. Figure 2 (left) shows an example for a permutation **536214**. Now that we are ready, we move on to the description of a decorated permutation. A decorated permutation is a permutation table where each dot is decorated by $①, ⊗, ⊘,$ or $⊙$. Figure 2 (left bottom) shows an illustration of a decorated permutation. One of the important properties of decorated permutations is shown below.

**Proposition 2.2.** *(See Proposition $8$ in [75].)* *There exists one-to-one correspondence between decorated permutations with decorations $\hat{\delta} \in \{①, ⊗, ⊘, ⊙\}^n$ and leveled permutrees with $\delta(\mathbf{T}) = \hat{\delta}$.*

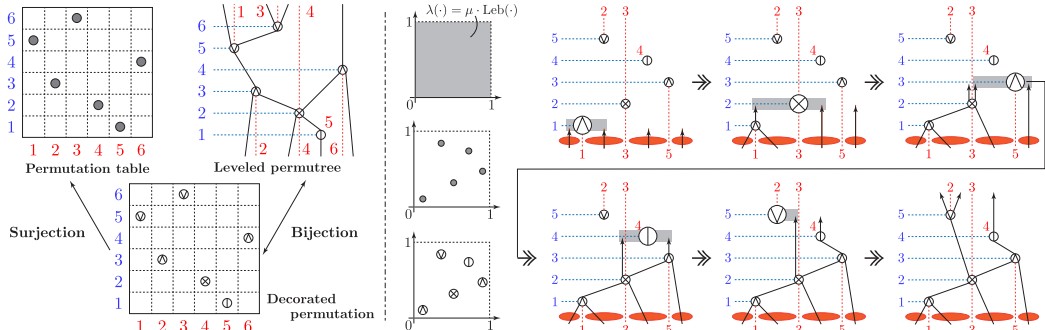

Figure 2: **Left: One-to-one correspondence between decorated permutations and leveled permutrees**. **Right: Permutree process as marked point process** - We introduce an intensity function $\lambda$ on the plane $[0,1] \times [0,1]$ (top left). Next, we generate random locations $\boldsymbol{l}_1, \dots, \boldsymbol{l}_n$ from the Poisson point process with intensity $\lambda$ (middle left). Then, for each random location, we independently assign one of the decorations $\{\oplus, \otimes, \oslash, \oslash\}$ from the categorical distribution as a random mark $m_i$ $(i = 1, \dots, n)$. By reading the positional relationship of the points as a permutation table, the resulting marked point $\{(\boldsymbol{l}_i, m_i) : i = 1, 2, \dots n\}$ can be converted to a *decorated permutation*. Furthermore, by the transformation used in Proposition 2.2 [75], the decorated permutation can be converted to a leveled permutree, as follows. First, we draw auxiliary lines (dashed lines colored red) below decorations $\otimes, \oslash$ and above decorations $\otimes, \oslash$. From this point on, we will stretch the permutree edges, and it is important to emphasize that the permutree edges do not cross these auxiliary lines. Next, focusing on the auxiliary lines extending to the bottom, we can view these as dividing the lower region into smaller subregions (indicated by the red ovals). The edges are then extended one by one from each subregion. As we extend the edges from the bottom to the top, when they reach the height of each vertex, we connect the adjacent edges to that vertex (indicated by the gray box). By doing this until all vertices are covered, we obtain a *leveled permutree*.

Now that we have reviewed the permutree findings, the next and subsequent sections will address three challenges: (i) How can we construct a stochastic process that can represent any permutree (in Section 3)? (ii) How can we construct a BNP prior model of the data using the stochastic process on the permutree (in Section 4)? (iii) What likelihood models can we combine the BNP prior with in actual machine learning applications (in Section 5)?

## 3 Permutree processes

The goal of this section is to construct a stochastic process that can represent any permutree; ideally, as is the basic philosophy of BNP, that stochastic process should also be able to simultaneously represent randomness with respect to complexity (in the context of permutrees, the number of vertices). In fact, our construction below can represent every permutree with an unlimited number of finite or infinite number of vertices in a unified manner, depending on certain hyperparameters. One thing to note in advance is that the stochastic process described in this section does not refer to any modeling of data. We will discuss data modeling in more detail in the next Section 4.

**Key insight** - Our strategy is to use point processes. Recall that, as discussed in Section 2, permutrees can be represented through leveled permutrees (surjection), and furthermore, leveled permutrees have a one-to-one correspondence (bijection) with decorated permutations (Proposition 2.2). Thanks to these facts, instead of dealing directly with permutrees (seemingly difficult to handle), we can obtain a model of permutrees indirectly by considering a model of decorated permutations. So how can we model decorated permutations? We represent the random decorated permutations as a *marked point process* by considering random permutations as a *point process* and random decorations as *marks*.

**Marked point process for decorated permutations** - We consider a marked point process consisting of a point process and associated marks, which can be expressed as $\{(\boldsymbol{l}_i, m_i) : i = 1, 2, \dots\}$, where $\boldsymbol{l}_1, \boldsymbol{l}_2, \dots$ are locations and $m_1, m_2, \dots$ are associated marks. Specifically, we employ the following Poisson process on a 2-dimensional plane $[0,1] \times [0,1]$ with discrete marks (Figure 2 right):

• **Random locations** - We draw the random locations $\boldsymbol{l}_1, \boldsymbol{l}_2, \dots$ from a Poisson point process on the plane $[0,1] \times [0,1]$ with the intensity function $\lambda : [0,1] \times [0,1] \to \mathbb{R}^+$, where $\mathbb{R}^+ = \{r : r > 0, r \in \mathbb{R}\}$. Although not essential, for the sake of simplicity, we use a *homogeneous* Poisson

point process, that is, $\lambda(A) = \mu \cdot \text{Leb}(A)$ for all measurable subset $A$ of $[0, 1] \times [0, 1]$, where $\text{Leb}(\cdot)$ indicates the Lebesgue measure, and $0 < \mu < \infty$ is a tunable variable. For convenience, let $\boldsymbol{l}_i = (l_{i,1}, l_{i,2})$, where $l_{i,1}$ and $l_{i,2}$ are the horizontal and vertical positions, respectively.

- **Random marks** - We draw the random marks $m_1, m_2, \ldots, m_n$ independently from a categorical distribution on $\{①, ⊗, ⓥ, ⌀\}$: $\text{Categorical}(c_①, c_⊗, c_ⓥ, c_⌀)$, where $c_* \geq 0$ $(* \in \{①, ⊗, ⓥ, ⌀\})$ denotes the probability that decoration $*$ is adopted.

**Transformation to leveled permutree** - The above marked point process can immediately lead to a random leveled permutree with the following procedure. Recall that, as discribed in Section 2, the leveled permutree is defined by (i) the decorations on the vertices $\mathbf{V}$ and (ii) the two bijective vertex labelings $p, q : \mathbf{V} \rightarrow [n]$. For the decoration of vertices, we consider the point set of the marked point process as the vertex set $\mathbf{V}$, and the mark $m_i$ assigned to the $i$-th point as the decoration of the $i$-th vertex $\mathbf{v}_i \in \mathbf{V}$. Thus, the remainder to be considered is the setting of two functions $p$ and $q$. By construction, we can obtain the indices $a_1, \ldots, a_n$ so that the random positions $\boldsymbol{l}_1, \ldots, \boldsymbol{l}_n$ are in ascending order in the horizontal direction, that is, $l_{a_1,1} < l_{a_2,1} < \cdots < l_{a_n,1}$ (Recall that $\boldsymbol{l}_i = (l_{i,1}, l_{i,2})$, and $l_{i,1}$ represents the horizontal position). Similarly, in the vertical direction, we can obtain the indices $b_1, \ldots, b_n$ so that $l_{b_1,2} < l_{b_2,2} < \cdots < l_{b_n,2}$. Now, if we choose to set $p(\mathbf{v}_{a_i}) = i$ and $q(\mathbf{v}_{b_i}) = i$ for $i = 1, 2, \ldots, n$, then $p$ and $q$ satisfy the requirement of bijective functions. By the above, we have seen that indeed the marked point process provides us with what we need to define a leveled permutree, that is, the vertex decorations and two bijective functions $p$ and $q$. Finally, Figure 2 (right) show the procedure for explicitly converting a marked point process to a random leveled permutree. Inheriting Proposition 2.2 and the result (with the proof procedure) of [75, Proposition 8], we can confirm that this transformation is well defined (See Appendix E for details).

# 4    Data modeling with permutree process

The purpose of this section is to show how the permutree process described earlier can be used for modeling actual data. More specifically, this consists of the following two issues:

- **How to represent data using permutrees**: As permutrees themselves are simply mathematical objects, we must be clear about how we relate them to data modeling and analysis. In fact, there are many possible ways to describe data by permutrees. We consider the situation where a *data path* (a lineage to describe the data in conjunction with some likelihood model, such as the evolutionary model in Section 5) from one of the lower terminal nodes to one of the upper terminal nodes on the permutree is assigned to each data (Figure 3 (a), top). For example, if we restrict the permutree to one of its special cases, the binary tree, this data path is attributed to the path from the root to the terminal node, which is a situation commonly used in hierarchical clustering (Figure 3 (a), bottom).[1] We show a strategy to represent this random data path using a special variant of the nested Chinese restaurant process [10].

- **How to "implement" a permutree process**: In the previous section, we have shown that a marked point process with an intensity function $\lambda : [0, 1] \times [0, 1] \rightarrow \mathbb{R}^+$ can be used for the stochastic process on permutrees. On the other hand, another important topic is to clarify how to implement models (or more practically, what intensity function $\lambda$ to use) suitable for data analysis. Our strategy is to use the analogy of the stick-breaking process [89] to represent the infinite number of marked points generated from the marked point process, which is the entity of the permutree process. This can be viewed as a special case of using beta intensity in the horizontal direction and uniform intensity in the vertical direction as the intensity function $\lambda$ of the permutree process.

Experts in the BNP field might remind themselves that there are many other strategy options for the above topics in the light of the various findings that have emerged in the history of the development of the BNP method over the last 20 years. We will, for the sake of space, summarize in Appendix D the various ideas and their respective advantages and disadvantages with respect to those historical findings, including whether it is possible to extend the conventional tool of the "ordered" Chinese restaurant process [77, 85] for random binary trees to random permutrees. The main body of this paper focuses on the most straighforward strategy.

---

[1] Some readers may wish to consider another typical situation: a path where the data starts at one of the terminal nodes and "stops at an interior vertex" of the permutree. This modification can be easily achieved by additionally introducing a chain of Bernoulli trials [10] or a time limit by representing the growth of the path as a Markov process on a virtual time axis [88, 87]. Therefore, we will focus on the most basic situation.

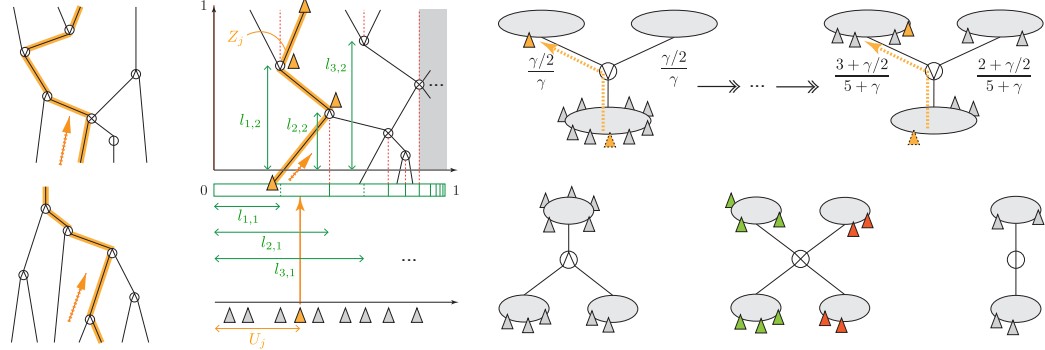

(a) Data path    (b) Marked stick-breaking process        (c) Two-table Chinese restaurant process

Figure 3: **(a) Data path** - We consider each data as having a data path (a lineage to describe the data in conjunction with some likelihood model) from one of the lower terminal nodes to one of the upper terminal nodes on the permutree (top). This will convince us of its generality and applicability, as it is attributed to the hierarchical clustering from one of the terminal nodes to the root when restricting the permutree to the special case of a binary tree (bottom). **(b) Marked stick breaking process** - Inspired by the stick-breaking representation [89] for the construction of Dirichlet processes [23], in order to represent a random permutree of infinite size, we can represent the random vertex positions of the permutree by the stick-breaking process in the horizontal direction and uniform random measures in the vertical direction. **(c) Two-table Chinese restaurant process** (Variant of two-class Dirichlet allocation) - The data allocated to the lower terminal nodes are successively merged and distributed depending on the mark of each vertex, according to the law of the 'the rich get richer', to select paths.

**Permutree of infinite size** - We first generate random positions $l_1, l_2, \dots, l_k = (l_{k,1}, l_{k,2}), \dots$ in the point process of the permutree process, as shown in Figure 3 (b), using the stick-breaking process:

$$\beta_k \sim \text{Beta}(1, \alpha), \quad l_{k,1} = \sum_{i=1}^{k} \left\{ \beta_i \prod_{i'=1}^{i-1} (1 - \beta_{i'}) \right\}, \quad l_{k,2} \sim \text{Uniform}([0, 1]), \quad (1)$$

where $\alpha > 0$ is the concentration parameter. As in the original permutree process, each point mark $m_k$ ($k = 1, 2, \dots$) is generated from a categorical distribution: $m_k \sim \text{Categorical}(c_{①}, c_{⊗}, c_{⑩}, c_{⑪})$. As mentioned earlier, by the procedure in Figure 2 (right), we can transform this sample (i.e., a set of infinite number of marked points) drawn from the permutree process into a uniquely single permutree.

**Data assignments to bottom terminal nodes** - Next, we can represent data modeling by the *paintbox* scheme for the random permutree generated from the marked stick-breaking process described earlier. We associate one uniform random variable $U_j$ for each data indexed by $j = 1, 2, \dots, N$ ($N \in \mathbb{N}$): $U_j \sim \text{Uniform}([0, 1])$. Similar to Kingman's representation to the exchangeable partitions, called *paintbox* schemes [44, 11], we choose which terminal node on the lower edge of the permutree to assign the $j$th data to, depending on which stick in the stick-breaking process this random variable $U_j$ is located on $[0, 1]$, as shown in Figure 3 (b).

**Data path modeling** - Finally, we model the path assignment for each data by choosing a path that starts at this assigned lower terminal node and reaches one of the upper terminal nodes through the following *two-table Chinese restaurant process* (i.e., variant of two-class Dirichlet allocation):

- ⑩ - We break up the set of data flowing in, following the left-right table-assignement operation below[2]: the first data is chosen uniformly at random from either the left or the right table. For the $n$th data, the left table is chosen with probability $(\mathcal{N}_{\text{Left}} + \gamma/2)/(n + \gamma)$ and the right table with probability $(\mathcal{N}_{\text{Right}} + \gamma/2)/(n + \gamma)$, where $\gamma > 0$ is a hyperparameter, and $\mathcal{N}_{\text{Left}}$ and $\mathcal{N}_{\text{Right}}$ are the number of data allocated so far to the left and right tables respectively.

- ⑪ - We merge the sets of data flowing from the two lower branches and feed them into the upper.

- ⊗ - It would be straightforward to perform operations whose marks are ⑩ and ⑪ together. Another promising option is the representation of data flowing from the left parent to the left child and from the right parent to the right child. This can be interpreted as giving the mark ⊗ the ability to *partitioning*. This interpretation also plays an important role in the validity of *finite truncation*, which will be discussed below.

---

[2]This is equivalent to a categorical-Dirichlet hierarchical model with two classes (two tables). We can obtain the form described in the text by marginalising the intermediate Dirichlet variable in this hierarchical model.

- ① - We pass on the whole data set that flows in, all the way to the top.

For notational simplicity, we will denote the random variable for the $j$th data path by $Z_j$. For a sample $z$ of data paths between the upper and lower terminal nodes of the permutree (specified by a sequence of edges), the above generative probabilistic model allows us to evaluate the probability $\mathbb{P}[Z_j = z]$ of the $j$th data choosing a data path sample $z$.

**Property #1: Exchangeability** - Random data paths based on the generative probability model described above have *exchangeability*, an important property common to most BNP models [3, 36, 41]. Simply put, the model probability is invariant to the indexing of the data. As a result, it follows the philosophy of BNP models that even if the actual data to be observed is finite, the model itself, with infinite complexity, can reflect the uncertainty due to unobserved data. More specifically, this can be summarised as the following statement:

**Proposition 4.1** (Exchangeability). *For any permutation $\sigma$ of length $N$ ($N \in \mathbb{N}$), we have $\mathbb{P}[Z_1 = z_1, Z_2 = z_2, \ldots, Z_N = z_N] = \mathbb{P}[Z_{\sigma(1)} = z_1, Z_{\sigma(2)} = z_2, \ldots, Z_{\sigma(N)} = z_N]$, where $z_j$ ($j \in [N]$) is a sample of paths of random permutrees. (See Appendix A.1 for proof.)*

**Property #2: Validity of finite truncation** - The above generative probability model requires in principle an infinite number of random variables for its description, but finite truncation works reasonably well for a finite number of actual observed data. This poses an inherently non-trivial challenge that is not present in the validity of approximating the stick-breaking process [89] for the Dirichlet process [23] with a finite number of stick-breaking procedures, which is a typical topic in the past [94, 87, 67]. The reason for this non-triviality is that the substructure of a permutree with infinite size is, in principle, affected by an infinite numnber of all marked vertices. Therefore, restricting the structure of the permutree to only some marked vertices may have a significant impact on the structure of the permutree. However, as the following statement shows, the substructure of the permutree has the good property that it depends only on a subset of marked vertices.

**Proposition 4.2** (Finite truncation). *In the above generative probability model of data indexed by $j = 1, 2, \ldots, N$ ($N \in \mathbb{N}$), we consider an event that all random variables $U_j$ ($j = 1, \ldots, N$), representing the horizontal position of the $j$th data, falls in the range $[0, 1 - \epsilon)$ as a situation with a sufficiently high probability $\mathbb{P}[\wedge_{j=1}^{N} 0 \leq U_j < 1 - \epsilon] = \prod_{j=1}^{N} \mathbb{P}[0 \leq U_j < 1 - \epsilon] > 1 - \epsilon \cdot \mathcal{O}(N)$, where $\epsilon > 0$ is a tiny real value. In this situation, there exists some natural number $K < \infty$, and all data paths are assigned with probability $1$ only to paths on the finite-size random permutree generated from the random marked points $l_1, l_2, \ldots, l_K$. (See Appendix A.2 for proof.)*

## 5  Application to phylogenetic permutree analysis

This section presents an application example of using the prior model representation of data using permutrees, which has been described in Section 4, in conjunction with a likelihood model in a specific application. One of the most promising applications of permutrees would be phylogenetic tree analysis for DNA molecular sequence data (e.g., CAGTC). DNA sequences from one or more populations are related by a branching structure known as genealogy. The complex correlative structure of a collection of DNA sequences can be represented as a phylogenetic tree, a record of *coalescence*, *recombination*, and *mutation* events in the history of the target organism: *coalescence* refers to the event in which two sequences are attributed to a common ancestor, *recombination* refers to the event in which a lineage splits into two sub-lineages when looking back in time from the present to the past, and *mutation* refers to the change of each letter of a DNA sequence over time.

**Challenges of conventional methods** - The most standard structure that has been used in phylogenetic analysis is the binary tree [66, 78, 58, 103, 98, 113, 107, 106, 60]. In fact, binary trees are very well suited to represent *coalescence* events in genealogy. However, one drawback of binary tree models is that they are not suitable for representing *recombination* events in a way that is compatible with coalescence events. To circumvent this drawback, the ancestral recombination graphs (ARGs) have sometimes been used as models that can represent both coalescence and recombination at the same time [52, 42, 80, 72, 90, 28]. However, it is not easy to model or infer ARGs directly, and often indirect ways of representing models by other perspectives (e.g., the fragmentation-coagulation process [92, 19]) have been explored, or approximate models (e.g., the coalescent hidden Markov model [34, 53] and the sequentially Markov coalescent model [80]) have been considered. Moreover, conventional phylogenetic tree analysis, including not only ARGs but also binary tree models,

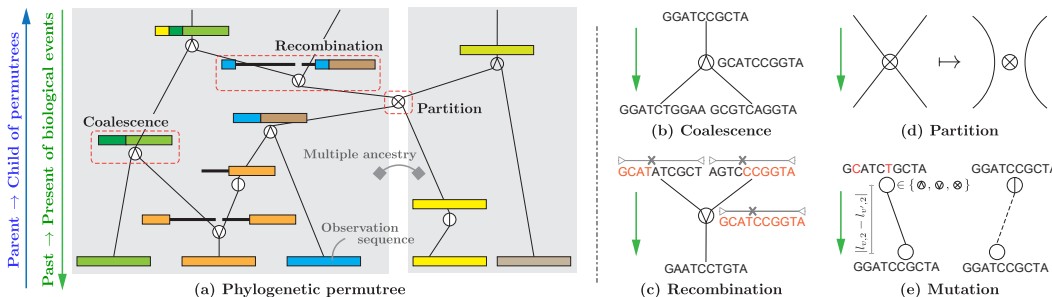

Figure 4: **(a) Phylogenetic permutree** can simultaneously and unifiedly represent **(b) coalescence**, **(c) recombination**, multiple ancestry through **(d) partition**, and **(e) mutation**. We note that the past (upper) to present (lower) direction (indicated by ↓) as biological events is the opposite of the parent (lower) to child (upper) direction (indicated by ↑) of the permutree as a purely mathematical object.

generally imposes a strong assumption that observed DNA sequences or observed taxa have a single ancestor. In other words, this implies that the inferred phylogenetic tree should be a strongly connected graph. Needless to say, such an assumption is reasonable for taxa that have been carefully selected by biologists. On the other hand, when we want to use a large number of taxa that are too large to be selected by experts as observation data (i.e., the situation that BNP methods are really aiming for), a mechanism that allows multiple ancestors to be inferred in a data-driven manner will be very useful. In light of the above, phylogenetic tree analysis requires a model that can represent coalescence, recombination, multiple ancestors, and mutation in a unified manner.

**Phylogenetic permutree** - As input observation data, we used DNA (molecular) sequences observed at letter length $S$ over $N$ species. For example, the sequence GAGTAC (i.e., $N = 1$ species) has length $S = 6$. We regard these DNA sequences as following a *phylogenetic permutree*. Specifically, we represent coalescence, recombination, multiple ancestry, and mutation events in genealogy by combining the four types of the decorations $\oslash, \obackslash, \otimes, \oplus$ with the following interpretations. We note that, to be consistent with the traditional notation of phylogenetic tree analysis, the past (upper) to present (lower) direction as biological events is the opposite of the parent (lower) to child (upper) direction of the permutree as a purely mathematical object that we have used in the diagrams so far.

- **Coalescence** $\oslash$ - A coalescence event represents two lineages (bottom side of Figure 4 (b)) having a common ancestral lineage (top side).

- **Recombination** $\obackslash$ - A recombination event represents the joining of two exclusive subsequences of two lineages (top side of Figure 4 (c)) by one lineage (bottom).

- **Partition** $\otimes$ - We give the decoration $\otimes$ the role of division so that a single permutree can represent a phylogenetic tree with multiple ancestors. Specifically, as shown in Figure 4 (d), we connect the two left edges and connect the two right edges resulting in two tree structures unconnected to each other on either side of decoration $\otimes$.

- **Backward in time** $\oplus$ (optional) - We assume that no mutation occurs while going back in time from a vertex to a vertex with $\oplus$ (Figure 4 (e)). This allows us to set the mutation rate in the evolutionary model as a single parameter common to all branches, and the mutation rate can be adjusted according to the permutree itself.

**Evolutionary models on permutrees** - Statistical models of gene mutation have a history of more than half a century, and a vast number of models have been proposed. An excellent recent review article can be found, for example, in [4]. For simplicity, we adopt two of the most popular models, the Jukes-Cantor model (JC) [40] and the generalized time reversible model (GTR) [91], for DNA sequences (i.e., words with A, G, C, and T as letters of the alphabet {A,G,C,T}, such as CCTAAG). JC is defined as a Markov process in which (1) all letters are independently generated from a uniform categorical distribution on {A,G,C,T} as initialization and (2) one letter (e.g., A) changes to another letter (e.g., G) after $t$ seconds with probability $(1-\exp(-4\alpha t))/4$ or does not mutate with probability $(1 + 3\exp(-4\alpha t))/4$, where $\alpha$ $(> 0)$ is a hyperparameter representing the mutation rate. Simply put, JC means that the transition probabilities of letters in mutation are fixed. GTR, on the other hand, can be regarded as a more flexible version of the JC model, in which the letter transition probabilities themselves are also estimated from the data as hidden parameters.

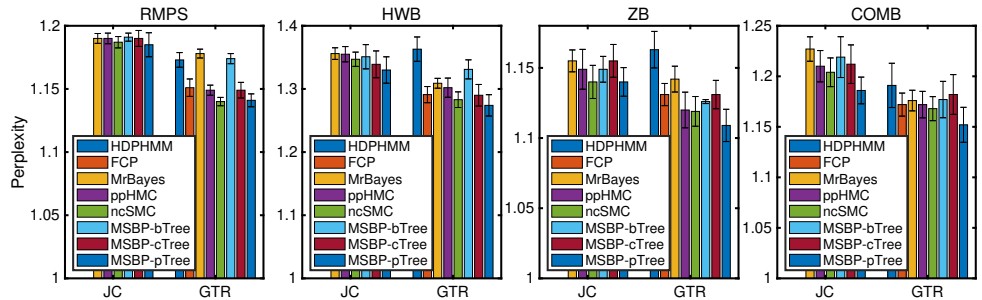

Figure 5: Experimental results of test perplexity (mean±std) comparison for real-world data.

**Demonstration** - We use the following three benchmark datasets [60] for DNA sequences: RMPS ($N = 64$ species, $S = 1008$ length) [86], HWB ($N = 41$, $S = 1137$) [33], and ZB ($N = 50$, $S = 1133$) [108]. In addition, to establish a situation where the permutree notion would be useful (i.e., multiple ancestry derived from exclusive disconnected graphs), we extract the sequences of these datasets by $S = 1000$ length from the beginning and mix them to create a dataset we call COMB ($N = 155$). We use the marked stick-breaking process (referred to as MSBP; Section 4) as our proposed model. Since MSBP can easily adjust the representational capabilities of of its own model, as ablation studies, we use MSBP-bTree as the one restricted to binary trees (with the prior $(c_{①}, c_{⊗}, c_{Ⓥ}, c_{Ⓐ}) \sim \mathrm{Dirichlet}(\epsilon/2, 0, 0, \epsilon/2)$), MSBP-cTree as the one restricted to Cambrian trees (with $\mathrm{Dirichlet}(\epsilon/3, 0, \epsilon/3, \epsilon/3)$), and MSBP-pTree as the main proposal permutrees (with $\mathrm{Dirichlet}(\epsilon/4, \epsilon/4, \epsilon/4, \epsilon/4)$), where we set $\epsilon = 0.01$. For the evolutionary model, we employ the mutation rate $\alpha \sim \mathrm{Gamma}(\epsilon', \epsilon')$, where $\epsilon' = 0.1$. We only present the case of $K = 100$ as the truncation level here, while we report the other cases in Appendix C. We compare these models to the hierarchical Dirichlet process hidden Markov model (HDPHMM) [8, 94, 5], the fragmentation-coagulation process (FCP) [92], and the binary tree model with the MrBayes [38, 46], the probabilistic path Hamilton Monte Carlo (ppHMC) [18], and the nested combinatorial sequential Monte Carlo (ncSMC) [60]. It is noted that HDPHMM and FCP do not use evolutionary models because they represent sequence data directly without tree structure. We held out $20\%$ letters of the input sequences for testing, and each model was trained using the remaining $80\%$ of the letters. Each inference method uses MCMC to estimate the posterior distribution by the following 100 samples: each method extracts 5 MCMC runs with different random numbers, and each MCMC run is sampled every 50 iterations after 2000 burn-in until 3000 iterations. We evaluate the models using perplexity as a criterion: $\mathrm{perplexity}(\cdot) = \exp(-(\log p(\cdot))/E)$, where $E$ is the number of missing letters in the input sequences. Figure 5 shows the comparison of the prediction performance of each method for the four sets of data. As an overall trend, it can be seen that the Cambrian tree and permutree models show better prediction performance than the binary tree model, which has limited expressive power.

## 6 Discussion and limitation

This paper (i) imports the notion of permutrees, recently invented in combinatorics, to Bayesian analysis, (ii) proposes the stochastic process that can represent various models such as permutations, trees, partitions, and factors in a unified manner, (iii) and applies it to phylogenetic permutree analysis.

**Limitations** - While our proposed permutree process can represent various combinatorial structures in a unified "prior model," the likelihood model that describes the data (as we have shown in the context of phylogenetic tree analysis in Section 5, for example) must be prepared separately by the user or engineer. Thus, while the permutree process is a tool that allows data-driven inference of the model structure as a broad framework, the design of the likelihood model needs to be carefully done manually. In the near future, the exploration of representing this likelihood model in some kind of black box function model would be an important research direction.

**Remaining challenge** - In the technical context of the BNP field, an important topic is whether a marginalized representation of the marked stick-breaking process, an infinite-dimensional intermediate random variable in the representation of data paths with exchangeability described in Section 4, can be obtained. This topic is a question closely related to the Aldous-Hoover-Kallenberg representation theorem for exchangeability in general [3, 36, 41]. As a more familiar analogy, it corresponds to the fact that if we marginalise the stick-breaking process representation in a Dirichlet process infinite mixture model, then we obtain the Chinese restaurant process representation. Our strategy and budding attempts on this question are summarized in Appendix D.

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

# A Properties of marked stick-breaking process

This section provides proofs of Propositions 4.1 and 4.2 concerning two properties of the marked stick-breaking process omitted in Section 4 of the main text.

## A.1 Exchangeability

One of the most important properties of random data paths drawn from the generative probabilistic model described in Section 4 is *exchangeability*, that is, the model probability is invariant to the indexing of the data. More specifically, this can be summarised as the following statement:

**Proposition A.1** (Exchangeability; Proposition 4.1)**.** *For any permutation $\sigma$ of length $N$ ($N \in \mathbb{N}$), we have $\mathbb{P}[Z_1 = z_1, Z_2 = z_2, \ldots, Z_N = z_N] = \mathbb{P}[Z_{\sigma(1)} = z_1, Z_{\sigma(2)} = z_2, \ldots, Z_{\sigma(N)} = z_N]$, where $z_j$ ($j \in [N]$) is a sample of paths of random permutrees.*

*Proof.* Broadly as a whole, we will check two following facts:

    (i) The random data assignments to bottom terminal nodes by the stick-breaking process [89] and the Kingman's paintbox scheme [44] are themselves exchangeable.

    (ii) The selection of data paths by the two-table Chinese restaurant process is exchangeable.

**Exchangeability of data assignments to bottom terminal nodes** - We denote the index of the stick of the stick-breaking process to which the $j$th ($j = 1, \ldots, N$) data is assigned by the random variable $Z_j^{\text{bottom}}$. It follows from the model construction that, given a random partition of $[0, 1]$ drawn from the stick-breaking process, the random variable $U_j$ ($j = 1, \ldots, N$) is independent. As a result, for any permutation $\sigma$ of length $N$, we have

$$\mathbb{P}\Big[Z_1^{(\text{bottom})} = s_N, \ldots, Z_N^{(\text{bottom})} = s_N\Big] = \prod_{j=1}^{N} \Big[Z_j^{(\text{bottom})} = s_j\Big] = \prod_{j=1}^{N} \Big[Z_{\sigma(j)}^{(\text{bottom})} = s_j\Big]$$
$$= \mathbb{P}\Big[Z_{\sigma(1)}^{(\text{bottom})} = s_N, \ldots, Z_{\sigma(N)}^{(\text{bottom})} = s_N\Big], \quad (2)$$

where $s_j$ ($j = 1, \ldots, N$) is a sample of stick indices ($\in \mathbb{N}$).

**Exchangeability of data path selection** - Given the assignment of data to the terminal nodes, the choice of data paths follows a chain of the two-table Chinese restaurant process (see Figure 3 (c) in the main text) according to the decoration of each inner vertex of the permutree. It should be noted that in the two-table Chinese restaurant process, the data paths are chosen deterministically when the decorations are ⬰,⊗, and ⊘. Therefore, we only need to focus on the case of table partitioning (Figure 3 (c), top) when the decoration is ⓥ. It immediately from the model construction that the probability of partitioning the data when the decoration is ⓥ is obtained as follows:

$$\mathbb{P}\Big[Z_1 = z_1, \ldots, Z_N = z_N \mid Z_1^{(\text{bottom})} = s_N, \ldots, Z_N^{(\text{bottom})} = s_N\Big]$$
$$= \prod_r \frac{\left\{(1 + \frac{\gamma}{2}) \cdots (\mathcal{N}_{\text{Left}}^{(r)} + \frac{\gamma}{2})\right\} \cdot \left\{(1 + \frac{\gamma}{2}) \cdots (\mathcal{N}_{\text{Right}}^{(r)} + \frac{\gamma}{2})\right\}}{(1 + \gamma)(2 + \gamma) \cdots (n^{(r)} + \gamma)}, \quad (3)$$

where the variable $s_j$ is the index of bottom terminal nodes (i.e., the stick index of the stick-breaking process on $[0,1]$) included in the path sample $z_j$, the variable $n^{(r)}$ represents the number of data flowing to the $r$th permutree vertex from the bottom in the vertical direction in the collection of data path samples $z_1, \ldots, z_N$, and $\mathcal{N}_{\text{Left}}^{(r)}$ and $\mathcal{N}_{\text{Right}}^{(r)}$ represent the number of data to be partitioned into the left and right tables at the $r$thth vertex (if the decoration at that vertex is $\otimes$), respectively. It is important to note that the probability of selecting this datapath depends only on the number of data in the division of the table at each vertex. That is, in other words, it does not depend on the index of the data as follows:

$$\mathbb{P}\Big[Z_1 = z_1, \ldots, Z_N = z_N \mid Z_1^{(\text{bottom})} = s_N, \ldots, Z_N^{(\text{bottom})} = s_N\Big]$$
$$= \mathbb{P}\Big[Z_{\sigma(1)} = z_1, \ldots, Z_{\sigma(N)} = z_N \mid Z_{\sigma(1)}^{(\text{bottom})} = s_N, \ldots, Z_{\sigma(N)}^{(\text{bottom})} = s_N\Big], \tag{4}$$

for any permutation $\sigma$ with length $N$. Thus, it can be checked that the selection of data paths is exchangeable. From Equations 2 and 4, we have completed our proof. $\qquad\square$

## A.2 Validity of finite truncation

The generative probability model (described in Section 4) requires in principle an infinite number of random variables for its description, but finite truncation works reasonably well for a finite number of actual observed data. More specifically, we can summarize this property as follows:

**Proposition A.2** (Finite truncation; Proposition 4.2). *In the generative probability model (described in Section 4) of data indexed by $j = 1, 2, \ldots, N$ ($N \in \mathbb{N}$), we consider an event that all random variables $U_j$ ($j = 1, \ldots, N$), representing the horizontal position of the $j$th data, falls in the range $[0, 1 - \epsilon)$ as a situation with a sufficiently high probability $\mathbb{P}[\wedge_{j=1}^N 0 \le U_j < 1 - \epsilon] = \prod_{j=1}^N \mathbb{P}[0 \le U_j < 1 - \epsilon] > 1 - \epsilon \cdot \mathcal{O}(N)$, where $\epsilon > 0$ is a tiny real value. In this situation, there exists some natural number $K < \infty$, and all data paths are assigned with probability 1 only to paths on the finite-size random permutree generated from the random marked points $\boldsymbol{l}_1, \boldsymbol{l}_2, \ldots, \boldsymbol{l}_K$.*

*Proof.* It follows from the construction that the uniformly random random random variables $U_j$ ($j = 1, \ldots, N$) are independent, so that the probability of an event for which all those random variables fall within the range $[0, 1^\epsilon)$ can be checked as follows.

$$\mathbb{P}\Big[ \wedge_{j=1}^N 0 \le U_j < 1 - \epsilon \Big] = \prod_{j=1}^N \mathbb{P}\Big[0 \le U_j < 1 - \epsilon\Big] = \big(1 - \epsilon\big)^N > 1 - \epsilon N. \tag{5}$$

Then, from the construction of the marked stick-breaking process on $[0, 1]$, since there are countably infinite number of marked points in the range $[1 - \epsilon, 1] \times [0, 1]$, the probability that there exists some natural number $K < \infty$ and the corresponding decoration of it is $\otimes$ is 1. That is, we have

$$\mathbb{P}\Big[K < \infty \ \wedge \ m_K = \otimes \ \wedge \ 1 - \epsilon \le l_{K,1} \le 1\Big] = 1. \tag{6}$$

From the construction of the two-table Chinese restaurant process (described in Section 4) and the permutree requirement (C2) (described in Section 2), the data path assigned to the 1st to $K$th bottom terminal nodes in the stick-breaking process never reaches the $(K + 1)$th and subsequent indexed permutree vertices. Therefore, each have a data path only on the edges of the finite permutree consisting of the 1st to $K$th marked vertices. From the above, we have completed the proof. $\qquad\square$

# B Relationship between permutree process and other stochastic processes

The purpose of this section is to provide additional information to help the reader better understand the characteristics of the permutree process as marked point process.

We clarify the relationship between the permutree process and other existing stochastic processes. Specifically, the permutree process can lead to the *uniform random permutations* and the *Mondrian process* as its special cases. These relationships can be derived immediately from the fact that each can be expressed as a Poisson process of some sort. We will discuss each of these in specific detail

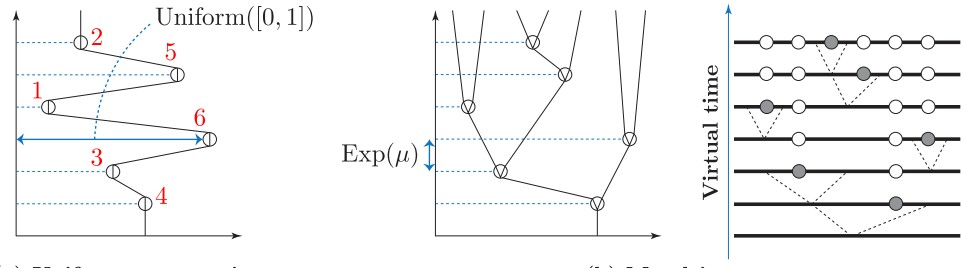

(a) Uniform permutation  (b) Mondrian process

Figure 6: Relationship between permutree process and other existing stochastic processes. **(a) Uniform random permutation** - If we restrict the decoration weights for marks to $(c_①, c_⊗, c_⑦, c_⊘) = (1, 0, 0, 0)$ and make the Poisson process homogeneous, the permutree process leads to a stochastic process that generates a *uniform random permutation*. This relation follows immediately from the following fact: If a collection of i.i.d. uniform random variables $U_1, U_2, \cdots \sim \text{Uniform}([0, 1])$ is ordered in ascending order, it follows a uniform random permutation. **(b) Mondrian process** - If we restrict the decoration weights to $(c_①, c_⊗, c_⑦, c_⊘) = (0, 0, 1, 0)$ and set $\lambda(\cdot) = \mu \cdot \text{Leb}(\cdot)$, the permutree process leads to a stochastic process that simulates a *Mondrian process* [88, 87] on $[0, 1]$ with the intensity $\mu$ and the budget 1. By viewing the vertical position in the marked point process as the moment when the event of the cut in the Markov process (i.e., the Mondrian process) occurs, and the horizontal position as the location where the cut occurs, the special permutation process described above can be reduced to a Mondrian process.

below. First of all, for self-containment, the core of the permutree process is restated, although it is the same as that detailed in the body of this paper.

**Permutree process** - We consider a marked point process consisting of a point process and associate marks, which can be expressed as $\{(\boldsymbol{l}_i, m_i) : i = 1, 2, \dots\}$, where $\boldsymbol{l}_1, \boldsymbol{l}_2, \dots$ are locations and $m_1, m_2, \dots$ are associated marks. Specifically, we employ the following Poisson process on a 2-dimensional plane $[0, 1] \times [0, 1]$ with discrete marks:

- **Random locations** - Draw the random locations $\boldsymbol{l}_1, \boldsymbol{l}_2, \dots$ from a Poisson point process on the plane $[0, 1] \times [0, 1]$ with the intensity function $\lambda : [0, 1] \times [0, 1] \rightarrow \mathbb{R}^+$, where $\mathbb{R}^+ = \{r : r > 0, r \in \mathbb{R}\}$. For notational convenience, we use $\boldsymbol{l}_i = (l_{i,1}, l_{i,2}) \ (\in \mathbb{R}^2)$, where $l_{i,1}$ and $l_{i,2}$ are the horizontal and vertical positions, respectively.

- **Random marks** - Draw the random marks $m_1, m_2, \dots, m_n$ independently from a categorical distribution on $\{①, ⊗, ⑦, ⊘\}$: $\text{Categorical}(c_①, c_⊗, c_⑦, c_⊘)$, where $c_* \ (* \in \{①, ⊗, ⑦, ⊘\})$ denotes the probability that decoration $*$ is adopted.

**Connection to uniform random permutation** - If we restrict the decoration weights for marks to $(c_①, c_⊗, c_⑦, c_⊘) = (1, 0, 0, 0)$ and make the Poisson process homogeneous (i.e., make the intensity function $\lambda$ uniform), the permutree process leads to a stochastic process that generates a *uniform random permutation*. This fact can be easily derived by interpreting the permutation process as follows. See also Figure 6 (a). By construction, we can obtain the indices $a_1, \dots, a_n$ so that the random positions $\boldsymbol{l}_1, \dots, \boldsymbol{l}_n$ are in ascending order in the horizontal direction, that is, $l_{a_1,1} < l_{a_2,1} < \cdots < l_{a_n,1}$. If we choose to set $p(\mathbf{v}_{a_i}) = i$ for the $i$-th vertex $\mathbf{v}_i \ (i = 1, 2, \dots)$ of the resulting permutree, then $p$ can lead to a permutation. The following fact shows that $p$ corresponds to a uniform random permutation:

**Proposition B.1.** *(See, for example, Lemma* 2.2 *in [32].) A uniform random permutation $\sigma$ with length $n$ can be obtained via a sequence of $n$ i.i.d.* $\text{Uniform}([0, 1])$ *random variables $W_1, \dots, W_n$ (Note that their values are distinct with probability 1), by taking $\sigma$ to be the unique permutation for which $W_{\sigma(1)} < \cdots < W_{\sigma(n)}$.*

**Connection to Mondrian process** - If we restrict the decoration weights to $(c_①, c_⊗, c_⑦, c_⊘) = (0, 0, 1, 0)$ and set $\lambda(\cdot) = \mu \cdot \text{Leb}(\cdot)$, the permutree process leads to a stochastic process that simulates a *Mondrian process* [88, 87] on $[0, 1]$ with the intensity $\mu$ and the budget 1. This fact can be easily derived by interpreting the permutation process as follows. In the above setup, the sample generated by the permutation process can be restricted to a binary tree by following the procedure described in

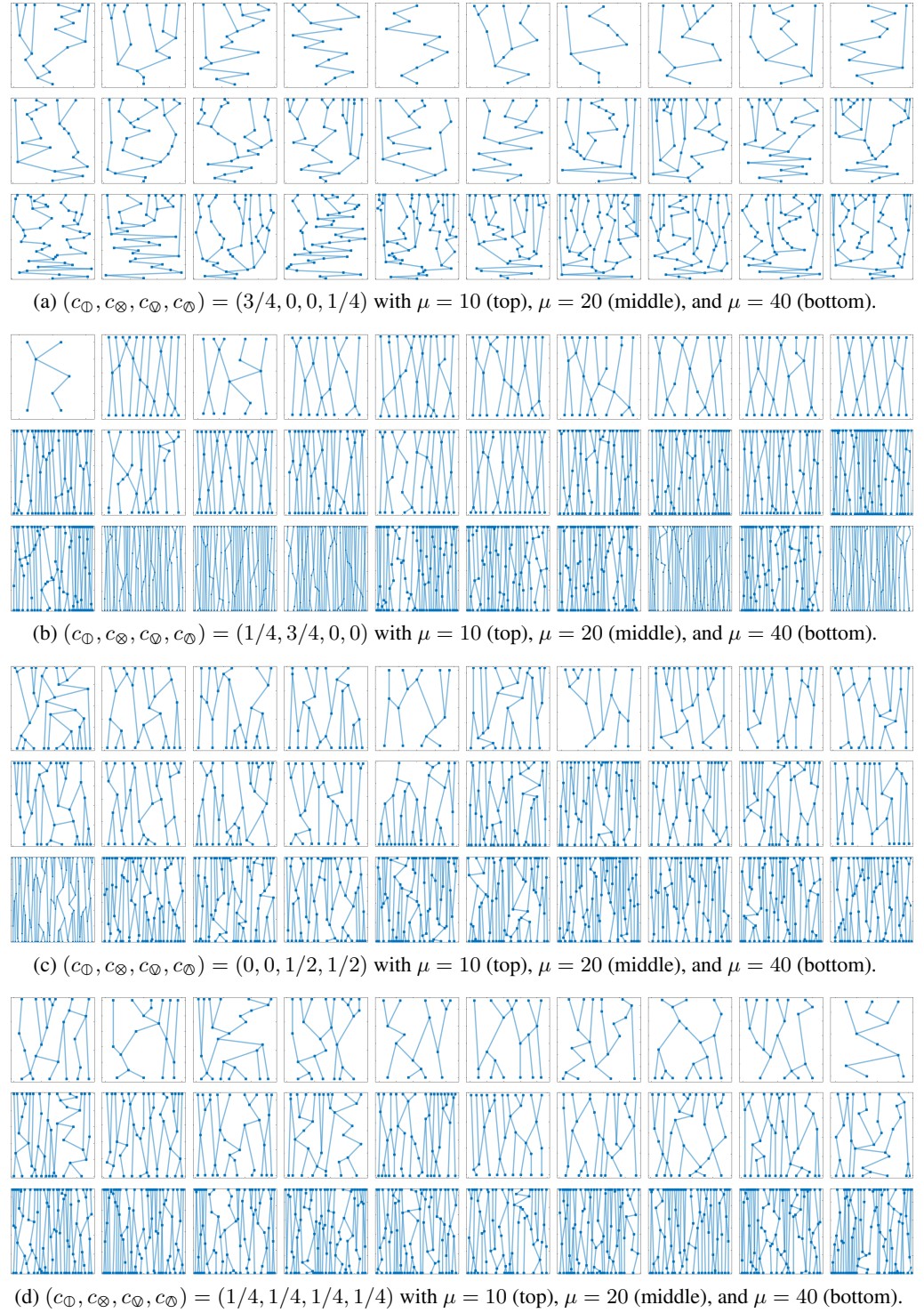

(a) $(c_\mathbb{O}, c_\otimes, c_\mathbb{V}, c_\mathbb{A}) = (3/4, 0, 0, 1/4)$ with $\mu = 10$ (top), $\mu = 20$ (middle), and $\mu = 40$ (bottom).

(b) $(c_\mathbb{O}, c_\otimes, c_\mathbb{V}, c_\mathbb{A}) = (1/4, 3/4, 0, 0)$ with $\mu = 10$ (top), $\mu = 20$ (middle), and $\mu = 40$ (bottom).

(c) $(c_\mathbb{O}, c_\otimes, c_\mathbb{V}, c_\mathbb{A}) = (0, 0, 1/2, 1/2)$ with $\mu = 10$ (top), $\mu = 20$ (middle), and $\mu = 40$ (bottom).

(d) $(c_\mathbb{O}, c_\otimes, c_\mathbb{V}, c_\mathbb{A}) = (1/4, 1/4, 1/4, 1/4)$ with $\mu = 10$ (top), $\mu = 20$ (middle), and $\mu = 40$ (bottom).

Figure 7: Samples drawn from permutree process with intensity $\lambda(\cdot) = \mu \cdot \mathrm{Leb}(\cdot)$ and decoration weights $(c_\mathbb{O}, c_\otimes, c_\mathbb{V}, c_\mathbb{A})$. Ten samples are generated for each parameter setting.

Section 3 (Figure 6) of the main text. From the fundamental properties of the Poisson process, the vertical interval between two adjacent vertices at random locations follows an exponential distribution $\mathrm{Exp}(\mu)$. Imagine the time evolution in the partition of a line segment of length 1 horizontally drawn from bottom to top, as shown in Figure 6 (b). The time evolution of this partition can be viewed as a

Markov process with an intensity $\mu$ and a time limit of 1. Furthermore, if we consider the horizontal location of the marked point process as the position where the line segment of length 1 is cut, we can consider this time evolution as a hierarchical partition of the line segment. Therefore, this can be regarded as a Mondrian process.

# C  Bayesian inference for phylogenetic permutree (omitted in Section 5)

This section reveals the Bayesian inference algorithm for phylogenetic permutree analysis using our permutree processes. For the sake of generality, we will use the marked point process representation described in Section 3 in particular as a permutree process. This argument can also be applied, with minor modifications, to the special case of the marked stick-breaking process (with its finite truncation) described in Section 4.

**Overview** - Standard Bayesian inference algorithms such as Markov chain Monte Carlo (MCMC) methods can be realized by sequentially iterating the following two update rules: (i) updating the permutree process (ii) updating the evolutionary model. Since the latter can be supported by standard inference methods to evolutionary models, it is the updating method of the former that is particularly important here. For the former, various inference algorithms that have been proposed for generic marked point processes and their extensions [68, 69, 26, 96, 43, 104, 30, 55, 31, 14, 35, 109] would be applicable, since the entity of the permutree process is a marked point process as shown in Section 3 of the main text. This section describes a useful inference method that exploits an important property of Poisson processes, namely, that a certain Poisson process can be obtained by *thinning* operations from another Poisson process with higher intensity. Section C.1 provides a brief description of the *thinning* operation for the Poisson process as a preliminary to our MCMC method. Then, Section C.2 once again writes down the whole generative probabilistic model, since it should be possible to see at a glance what the parameters to be inferred are in the permutree process and its phylogenetic tree described in Section 5. Finally, Section C.3 describes the MCMC inference algorithm.

## C.1  Preliminaries: thining operations for Poisson processes

Our MCMC method uses important properties of Poisson processes. Specifically, we will discuss how to represent a certain Poisson process via another Poisson process with higher intensity.

**Homogeneous Poisson process** - In this paper, we mainly consider *homogeneous* Poisson processes on $[0,1] \times [0,1]$, i.e., where the intensity function is given by a constant. A Poisson process on $[0,1] \times [0,1]$ with intensity $\mu$ (where $0 < \mu < \infty$) is a stochastic process for a random set of points, where the number of points belonging to $(x_1, x_2] \times (y_1, y_2]$ follows a Poisson distribution with the parameter $\mu(x_2 - x_1)(y_2 - y_1)$ for any $0 \leq x_1 < x_2 \leq 1$, $0 \leq y_1 < y_2 \leq 1$. For notational simplicity, we will denote a Poisson process on $[0,1] \times [0,1]$ with intensity $\mu$ by $\mathrm{PP}(\mu, [0,1] \times [0,1])$. We also recall that in the main text, we defined this homogeneous Poisson process as the Poisson process with the intensity function $\lambda(\cdot) = \mu \cdot \mathrm{Leb}(\cdot)$ as an equivalent expression, where $\mathrm{Leb}(\cdot)$ indicates the Lebesgue measure.

**Thinning operation on Poisson processes** - One of the most interesting properties of Poisson processes is that a Poisson process with a certain intensity can be obtained by applying the *thinning* operation from a Poisson process with a higher intensity. More specifically, a Poisson process with intensity $\mu$ can be constructed as follows:

    (i) We generate a random set of points from a Poisson process with intensity $\nu$ $(> \mu)$.

    (ii) For each point generated, independently decide whether or not to accept it with probability $\mu/\nu$.

    (iii) The set of only accepted points can be regarded as following the Poisson process with intensity $\mu$.

## C.2  Full description of phylogenetic permutree with permutree process

As input observation data, we consider DNA (molecular) sequences $\boldsymbol{x}_j$ $(j = 1, \ldots, N)$ observed at letter length $S$ over $N$ species. For example, the sequence $\boldsymbol{x}_j = \mathsf{GAGTAC}$ has length $S = 6$. Figure 8 (top) shows the observation DNA sequences as an $S \times N$ matrix. The four colors represented

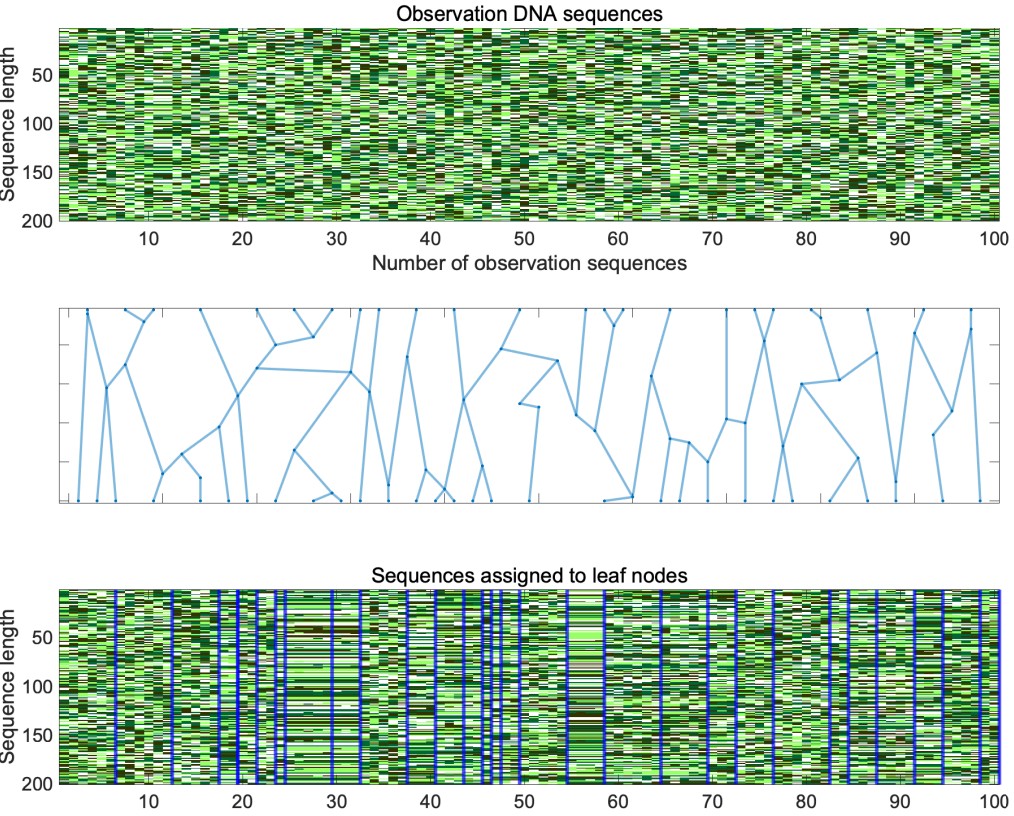

Figure 8: Observed DNA sequences (top), phylogenetic permutree (middle), and observed DNA serquences assigned to leaf nodes of phylogenetic permutree (bottom). Note that each leaf node (lower) of the phylogenetic permutree does not necessarily have to be the assignment of a single observation DNA sequence. The blue dividing line in the figure below represents a group of DNA sequences where each parcel corresponds to one leaf (lower) node. The phenomenon that each observed sequences within the same group is different is due to mutation events based on the evolutionary model.

by each element of the matrix correspond to the four different letters A,C,G, and T. We regard these DNA sequences as following a phylogenetic tree based on a permutree. First, we generate the marked points $\{(l_i, m_i) : i = 1, 2, \ldots n\}$ and the corresponding permutree $\mathbf{T}$ from the permutree process. We recall that the transformation from marked points to permutree can be performed by the transformation in Figure 9 (which we will call MPP2PT). Then, we represent coalescence, recombination, multiple ancestry, and mutation events in genealogy by combining the four types of the decorations $\oslash, \oslash, \otimes, \oplus$ with the following interpretations:

- **Coalescence** $\oslash$ - A coalescence event represents two lineages having a common ancestral lineage.

- **Recombination** $\oslash$ - A recombination event represents the joining of two exclusive subsequences of two lineages by one lineage.

- **Partition** $\otimes$ - We give the decoration $\otimes$ the role of division so that a single permutree can represent a phylogenetic tree with multiple ancestors. Specifically, we connect the two left edges and similarly connect the two right edges to lead to two unconnected tree structures on either side of decoration $\otimes$.

- **Backward in time** $\oplus$ (optional) - We suppose that no mutation occurs when going back in time from a vertex to a vertex with $\oplus$.

Figure 10 shows an intuitive illustration of the above interpretation of the transformation from a permutree $\mathbf{T}$ to a phylogenetic permutree $\mathcal{T}$. We will refer to this transformation as PT2PP : $\mathbf{T} \mapsto \mathcal{T}$. Here, we will use the vertical coordinate $l_{i,2}$ of each marked point as a representation of how far back in time each vertex is in the phylogenetic tree. Each vertex $v$ of the phylogenetic tree $\mathcal{T}$ shall have a hidden DNA sequence $\boldsymbol{h}_v$ (i.e., a sequence of length $S$ with each element having the letter from A,C,G, and T), which shall mutate according to the gene evolutionary models, such as the Jukes-Cantor model (JC) [40] and the generalized time reversible model (GTR) [91]. Figures 11 and 12 show examples of the evolution of the hidden DNA sequences $(\boldsymbol{h}_v)_{v \in \mathcal{T}}$ (e.g., sequence length $S = 10$) on the phylogenetic tree $\mathcal{T}$ in the mutation-prone and mutation-resistant cases, respectively. For notational convenience, we will denote the gene evolutionary models with mutation rate $\alpha$ $(> 0)$ on the phylogenetic tree $\mathcal{T}$ and mark locations $\boldsymbol{l}_1, \ldots, \boldsymbol{l}_n$ by $\mathrm{Evo}(\mathcal{T}, (\boldsymbol{l}_i)_{i=1}^n, \alpha)$. Finally, each of the $N$ input sequences is independently assigned to a data path from the two-table Chinese restaurant process (refered to as 2tCRP) with the concentration parameter $\gamma > 0$. In short, the overall model can be summarized as follows:

$$\boldsymbol{l}_1, \boldsymbol{l}_2, \ldots \boldsymbol{l}_n \sim \mathrm{PP}(\mu, [0,1] \times [0,1]) \qquad : \textit{Locations} \qquad (7)$$

$$(c_{\oplus}, c_{\otimes}, c_{\varovee}, c_{\varowedge}) \sim \mathrm{Dirichlet}(\epsilon/4, \epsilon/4, \epsilon/4, \epsilon/4) \qquad : \textit{Decoration weights} \qquad (8)$$

$$m_i \sim \mathrm{Categorical}(c_{\oplus}, c_{\otimes}, c_{\varovee}, c_{\varowedge}) \qquad : \textit{Marks} \qquad (9)$$

$$\mathbf{T} \leftarrow \mathrm{MPP2PT}((\boldsymbol{l}_i, m_i)_{i=1}^n) \qquad : \textit{Permutree} \qquad (10)$$

$$\mathcal{T} \leftarrow \mathrm{PT2PP}(\mathbf{T}) \qquad : \textit{Phylogenetic permutree} \qquad (11)$$

$$\alpha \sim \mathrm{Gamma}(\epsilon', \epsilon') \qquad : \textit{Mutation Rate} \qquad (12)$$

$$(\boldsymbol{h}_v)_{v \in \mathcal{T}} \sim \mathrm{Evo}(\mathcal{T}, (\boldsymbol{l}_i)_{i=1}^n, \alpha) \qquad : \textit{DNA evolution} \qquad (13)$$

$$Z_1, \ldots, Z_N \sim \mathrm{2tCRP}(\gamma) \qquad : \textit{Data paths} \qquad (14)$$

$$\boldsymbol{x}_j \sim \mathrm{Evo}(\mathcal{T}|_{Z_j}, \boldsymbol{l}_{Z_j}, \alpha) \qquad : \textit{Observation sequence} \qquad (15)$$

for $i = 1, 2, \ldots, n$ and $j = 1, \ldots, N$, where $\mathcal{T}|_{Z_j}$ refers to a phylogenetic tree (a tree consisting of one edge and two vertices at either end) from which only the leaf nodes and their children are extracted from the phylogenetic tree $\mathcal{T}$. Since the variables $\epsilon$ and $\epsilon'$ are hyperparameters for the non-informative prior distributions, it is standard to use them fixed to tiny values. Can we then consider how to directly infer the above generative probability model? Certainly, it is possible in principle to infer the above generative probability model as it is by direct updating of the permutree process as shown in Figure 13. However, in such direct inference, the complexity $n$ is often strongly affected by bad local modes, and often the Markov chain is entangled in the local optima, resulting in slow convergence. Therefore, using the properties of Poisson processes described in the preparation, a method can be considered to reduce the influence of such local optima by taking the dare to have redundant model parameters. Equation (7) and (9) can be rewritten as follows.

$$\hat{\boldsymbol{l}}_1, \hat{\boldsymbol{l}}_2, \ldots, \hat{\boldsymbol{l}}_K \sim \mathrm{PP}(\mu, [0,1] \times [0,1]) \qquad : \textit{Redundant locations} \qquad (16)$$

$$\hat{m}_i \sim \mathrm{Categorical}(c_{\oplus}, c_{\otimes}, c_{\varovee}, c_{\varowedge}) \qquad : \textit{Redundant marks} \qquad (17)$$

$$b_i \sim \mathrm{Bernoulli}(\mu/\nu) \qquad : \textit{Binary indicators} \qquad (18)$$

$$\boldsymbol{l}_1, \boldsymbol{l}_2, \ldots \boldsymbol{l}_n \leftarrow \left\{ \hat{\boldsymbol{l}}_1, \ldots, \hat{\boldsymbol{l}}_K \mid b_i = 1 \ (i = 1, \ldots, K) \right\} \qquad : \textit{Locations} \qquad (19)$$

$$m_1, m_2, \ldots m_n \leftarrow \{ \hat{m}_1, \ldots, \hat{m}_K \mid b_i = 1 \ (i = 1, \ldots, K) \} \qquad : \textit{Marks} \qquad (20)$$

for $i = 1, \ldots, K$. The above is the full phylogenetic tree model based on the permutree. One point to recall here is that permutrees includes binary trees and Cambrian trees as special cases (as discussed in Remark 2.1 of the main text). Therefore, the permutree process can be attributed to various models by adjusting the prior distribution for the decoration weights $(c_{\oplus}, c_{\otimes}, c_{\varovee}, c_{\varowedge})$. Figure 14 (left) shows the permutree process with $(c_{\oplus}, c_{\otimes}, c_{\varovee}, c_{\varowedge}) \sim \mathrm{Dirichlet}(\epsilon/2, 0, 0, \epsilon/2)$ as BINARYTREE (restricting the expressive power to binary trees). Figure 14 (right) shows $(c_{\oplus}, c_{\otimes}, c_{\varovee}, c_{\varowedge}) \sim \mathrm{Dirichlet}(\epsilon/3, 0, \epsilon/3, \epsilon/3)$ as CAMBRIANTREE (restricting it to Cambrian trees).

**Joint probability** - For notational convenience, we use $\boldsymbol{X} := (\boldsymbol{x}_1, \ldots, \boldsymbol{x}_N)$, $\boldsymbol{H} := (\boldsymbol{h}_v)_{v \in \mathcal{T}}$, $\boldsymbol{Z} := (Z_1, \ldots, Z_N)$, $\boldsymbol{b} := (b_1, \ldots, b_K)$, $\hat{\boldsymbol{L}} := (\hat{\boldsymbol{l}}_1, \ldots, \hat{\boldsymbol{l}}_K)$, $\hat{\boldsymbol{m}} := (\hat{m}_1, \ldots, \hat{m}_K)$, and $\boldsymbol{c} =$

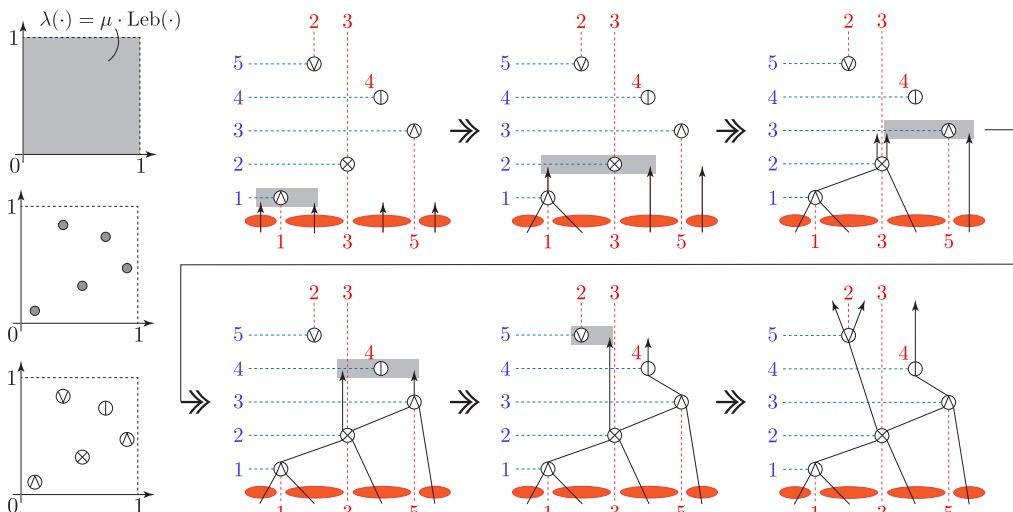

Figure 9: (Reprinted from the main text.) **Left: Marked point process** - We introduce an intensity function $\lambda$ (e.g., uniform measure) on the plane $[0, 1] \times [0, 1]$ (top figure). Then, we generate random locations $\boldsymbol{l}_1, \ldots, \boldsymbol{l}_n$ from the Poisson point process of the intensity $\lambda$ (middle figure). Finally, for each random location, we independently assign one of the decorations $\{①, ⊗, ⓥ, ⓐ\}$ from the uniform categorical distribution as a random mark $m_i$ ($i = 1, \ldots, n$) (bottom figure). The resulting marked point process $\{(\boldsymbol{l}_i, m_i) : i = 1, 2, \ldots n\}$ can be regarded as a random *decorated permutation*. **Right: Transformation to random permutree** - Note that the marked points generated from the marked point process can be considered as a decorated permutation by noting its horizontal and vertical ordering. Since decorated permutations and leveled permutrees have a one-to-one correspondence, we are guaranteed to be able to construct their bijective transformation. First, auxiliary lines (red dashed lines) are drawn below decorations $⊗$, $ⓐ$ and above decorations $⊗$, $ⓥ$. From this point on, we will extend the permutree edges, but it is important to emphasize that the permutree edges do not cross these auxiliary lines. Next, if we look at the auxiliary lines extending all the way to the bottom, we can see that this divides the lower region into smaller sub-regions (indicated by the red oval). Then, one edge is extended from each sub-region. The edges are extended from bottom to top, and when the height of each vertex is reached, the edges adjacent to that vertex are connected (indicated by the gray boxes). This is done until all vertices are covered, resulting in a *leveled permutree*. Finally, if we forget about the vertical position of each vertex in the leveled permutree and focus only on its structure as a directed tree, we obtain the corresponding *permutree*.

822    $(c_①, c_⊗, c_ⓥ, c_ⓐ)$. We obtain the following joint probability density function:

$$P_{\text{joint}}(\boldsymbol{X}, \boldsymbol{H}, \boldsymbol{b}, \boldsymbol{Z}, \hat{\boldsymbol{L}}, \hat{\boldsymbol{m}}, \boldsymbol{c}) = P_{\text{obs}}\left(\boldsymbol{X}; \hat{\boldsymbol{L}}, \hat{\boldsymbol{m}}, \boldsymbol{b}, \boldsymbol{z}, \alpha\right) \cdot P_{\text{evo}}\left(\boldsymbol{H}; \hat{\boldsymbol{L}}, \hat{\boldsymbol{m}}, \boldsymbol{b}, \alpha\right)$$

$$P_{\text{PP}}\left(\hat{\boldsymbol{L}}; \nu\right) \cdot P_{\text{Bernoulli}}(\boldsymbol{b}; \mu/\nu) \cdot P_{\text{Categorical}}\left(\hat{\boldsymbol{m}}; \boldsymbol{c}\right)$$

$$\cdot P_{\text{Dirichlet}}\left(\boldsymbol{c}; \epsilon\right) \cdot P_{\text{Gamma}}\left(\alpha; \epsilon'\right) \cdot P_{\text{2tCRP}}\left(\boldsymbol{Z}; \gamma\right), \tag{21}$$

823    where $P_{\text{obs}}$ is the probability density function (PDF) of Equation (15), $P_{\text{evo}}$ is PDF of Equation (13),
824    $P_{\text{PP}}$ is PDF of Equation (16), and subsequent terms are PDFs of the standard distributions. The
825    posterior distribution of the parameters $\boldsymbol{H}, \boldsymbol{b}, \boldsymbol{w}, \boldsymbol{z}, \hat{\boldsymbol{L}}, \hat{\boldsymbol{m}}, \boldsymbol{c}$ to be estimated is proportional to this
826    joint probability density.

827    ## C.3    Bayesian inference algorithm for phylogenetic permutree

828    We can construct the MCMC algorithm by iteratively repeating the following update rules for the
829    DNA evolution $\boldsymbol{H}$ on the phylogenetic permutree, the binary indicators $\boldsymbol{b}$, the leaf node weights $\boldsymbol{w}$,
830    the observation assignments $\boldsymbol{z}$, the redundant locations $\hat{\boldsymbol{L}}$, the redundant marks $\hat{\boldsymbol{m}}$, and the decoration
831    weights $\boldsymbol{c}$.

832    **Update rule for DNA evolution $\boldsymbol{H}$** - We recall that each element of the matrix $\boldsymbol{H} = (H_{s,j})_{S \times N}$
833    consists of one of the letters A, C, G, or T. Using Equation (21), we calculate the joint probability that

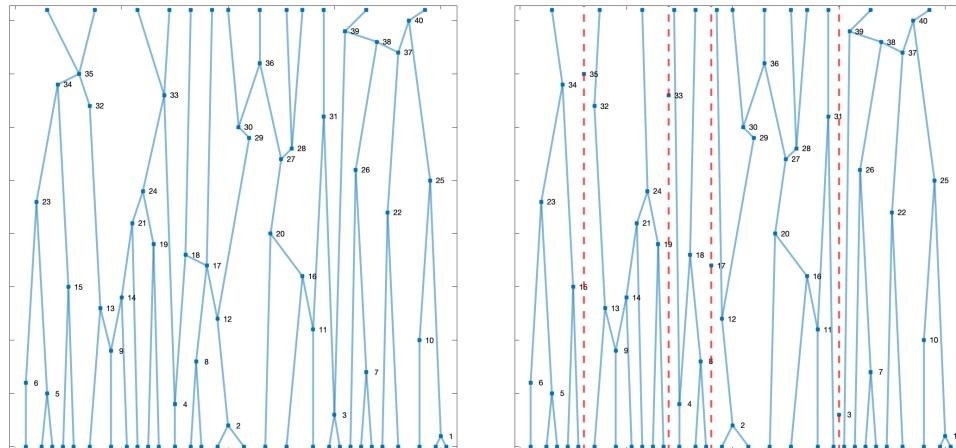

Figure 10: Intuitive illustration of transformation PT2PP from permutree (**left**) to phylogenetic permutree (**right**). The number assigned to each vertex **v** represents the function $q(\mathbf{v})$ (i.e., the order of the vertices vertically from bottom to top). We can regard this transformation as giving the role of the partition (red dotted line in the right figure) to the decoration $\otimes$ (i.e., the vertex with two parents and two children in the left figure).

each element $H_{s,j}$ is A, C, G, or T, respectively, and let $p_A$, $p_C$, $p_G$, or $p_T$ denote them respectively. Then we obtain the following Gibbs update rule:

$$H_{s,j} \sim \text{Categorical}(p_A, p_C, p_G, p_T) \qquad (s = 1, \ldots, S, \text{and } j = 1, \ldots, N) \qquad (22)$$

**Update rule for binary indicators $b$** - For each $i = 1, \ldots, K$, we can obtain the Gibbs update rule derived by calculating the posterior probability ratio for $b_i = 0$ and $b_i = 1$ using Equation (21). Specifically, we suppose that the value of the joint density for $b_i = 0$ is $\pi_0$ and the value for $b_i = 1$ is $\pi_1$, and then we obtain the following update rule:

$$b_i \sim \text{Bernoulli}\left(\pi_1 / (\pi_0 + \pi_1)\right) \qquad (i = 1, \ldots, K). \qquad (23)$$

**Update rule for leaf node weights $w$** - From the conjugacy of the Dirichlet and Categorical distributions, we obtain the following Gibbs update rule:

$$\left(w_1 \ldots, w_{|\mathcal{LN}(\mathcal{T})|}\right) \sim \text{Dirichlet}\left(\mathcal{N}_1 + \epsilon'', \ldots, \mathcal{N}_{|\mathcal{LN}(\mathcal{T})|} + \epsilon''\right) \qquad (i = 1, \ldots, K), \qquad (24)$$

where $\mathcal{N}_i$ $(i = 1, \ldots, |\mathcal{LN}(\mathcal{T})|)$ indicates the number of the observation sequences $\boldsymbol{x}_j$ $(j = 1, \ldots, N)$ which is assigned to the $i$th leaf node of the phylogenetic permutree $\mathcal{T}$.

**Update rule for observation assignments $z$** - Using Equation (21), we calculate the joint probability that each observation sequence $\boldsymbol{x}_j$ $(j = 1, \ldots, N)$ is assigned to the $i$th $(i = 1, \ldots, |\mathcal{LN}(\mathcal{T})|)$ leaf node of the phylogenetic permutree $\mathcal{T}$, and let $\bar{w}_i$ denote it. Then we obtain the following Gibbs update rule:

$$z_j \sim \text{Categorical}\left(\bar{w}_1, \ldots, \bar{w}_{|\mathcal{LN}(\mathcal{T})|}\right) \qquad (j = 1, \ldots, N) \qquad (25)$$

**Update rule for redundant locations $\hat{L}$** - We use the simple Metropolis-Hastings (MH) method. For each position, we generate a new candidate sample from the normalized probability measure $\hat{\lambda}$ of the intensity function $\lambda$, that is, $\hat{\boldsymbol{l}}_{\boldsymbol{i}} \sim \hat{\lambda}$ $(i = 1, \ldots, K)$, and decide whether to adopt it through the MH acceptance/rejection scheme using Equation (21).

**Update rule for redundant marks $\hat{m}$** - Using Equation (21), we calculate the joint probability that each element $\hat{m}_i$ has $\oslash, \varoslash, \otimes, \oplus$, respectively, and let $p_\oslash, p_\varoslash, p_\otimes, p_\oplus$ denote them respectively. Then we obtain the Gibbs update rule:

$$\hat{m}_i \sim \text{Categorical}\left(p_\oslash, p_\varoslash, p_\otimes, p_\oplus\right) \qquad (i = 1, \ldots, K) \qquad (26)$$

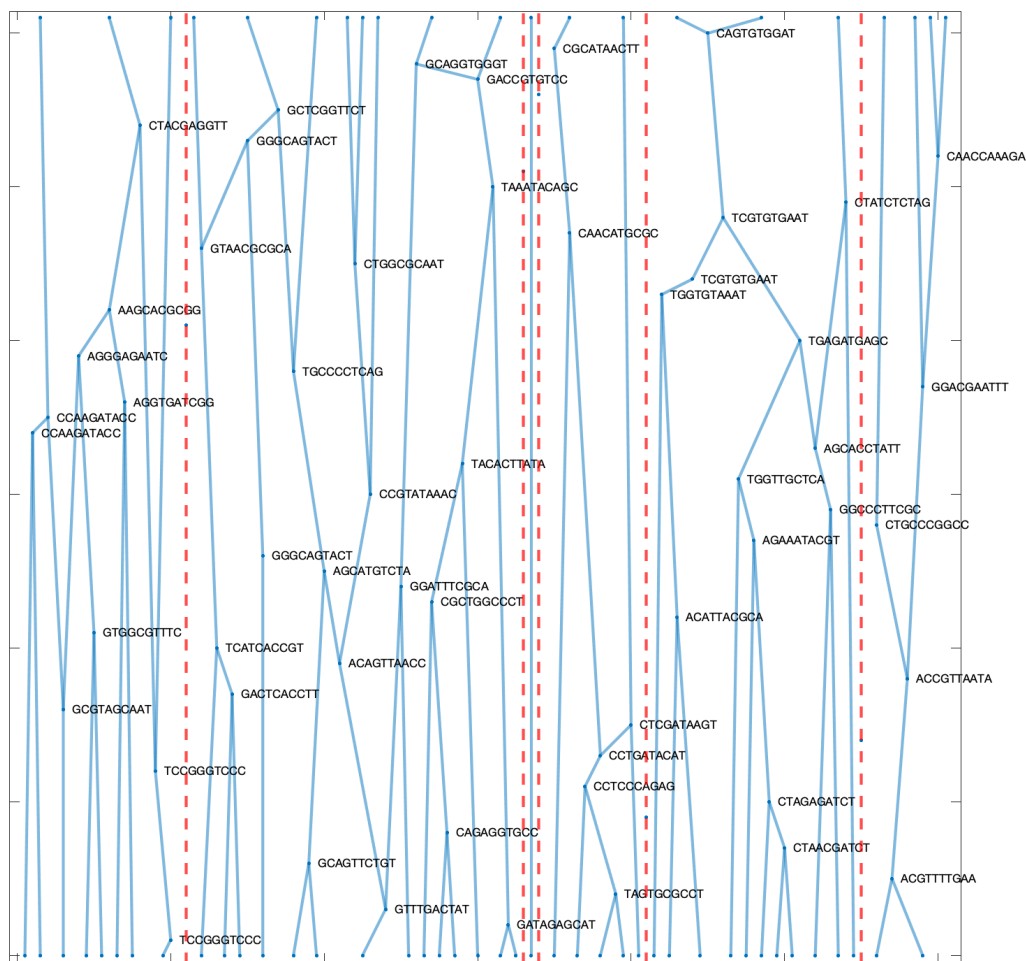

Figure 11: Phylogenetic trees and DNA evolution through Jukes-Cantor evolutionary model with mutation rate $\alpha = 0.1$.

**Update rule for decoration weights $c$** - From the conjugacy of the Dirichlet and Categorical distributions, we obtain the following Gibbs update rule:

$$(c_{\oslash}, c_{\obackslash}, c_{\otimes}, c_{\oplus}) \sim \text{Dirichlet}\Big(\mathcal{N}_{\oslash} + \epsilon/4, \mathcal{N}_{\obackslash} + \epsilon/4, \mathcal{N}_{\otimes} + \epsilon/4, \mathcal{N}_{\oplus} + \epsilon/4\Big), \qquad (27)$$

where $\mathcal{N}_*$ $(* \in \{\oslash, \obackslash, \otimes, \oplus\})$ indicates the number of the marks $\hat{m}_i$ $(i = 1, \ldots, K)$ which has the decoration $*$.

## C.4 Empirical impact of finite truncation

To investigate the empirical impact of finite censoring on the marked stick-breaking process described in Section 4, we report in Figure 15 the prediction performance for different levels of finite censoring, $K = 25, 50, 100$ and $150$, in the same experimental setup as in the main text (Section 5). It can be seen that when the level of finite truncations is extremely restricted, the prediction performance has been reduced, while when some level of censoring is ensured, the prediction performance is not so reduced. This can be seen as reflecting the fact that the marked stick-breaking process can adjust its own real model complexity in a data-driven manner according to the observation data.

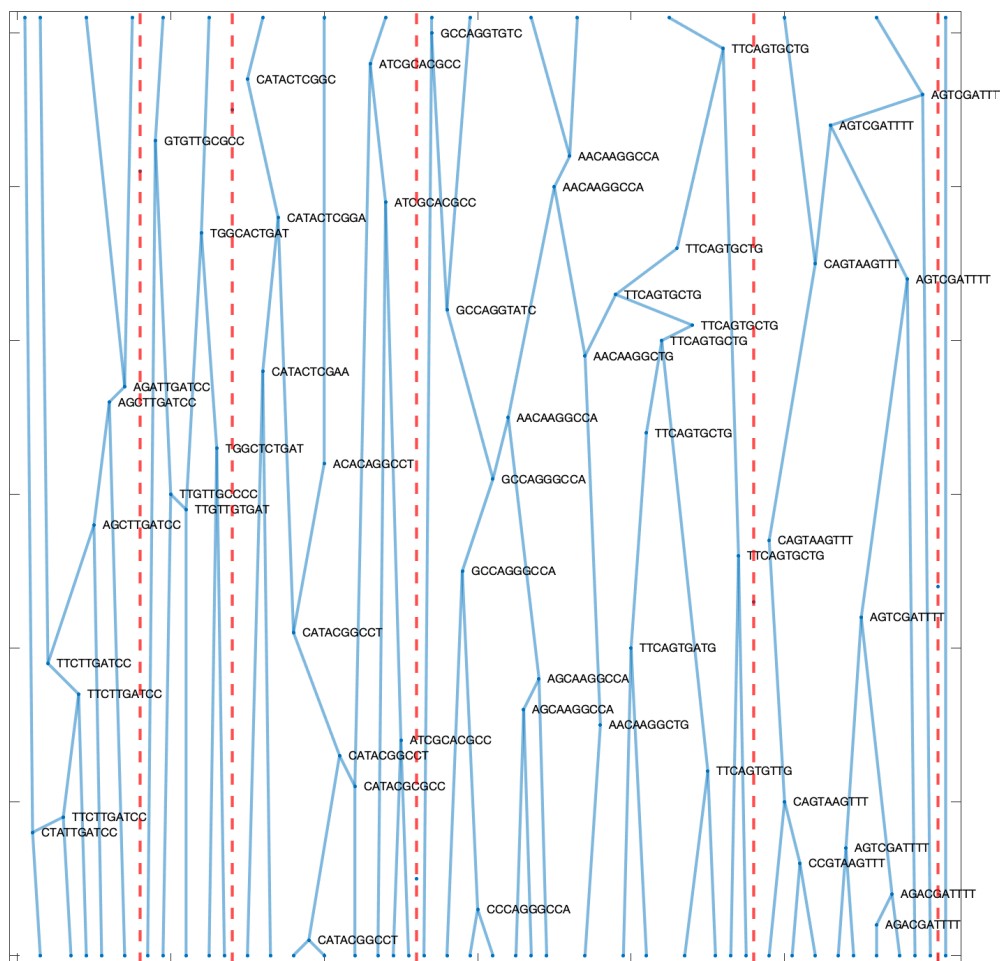

Figure 12: Phylogenetic trees and evolution of DNA lineages through Jukes-Cantor evolutionary model with mutation rate $\alpha = 0.001$ (i.e., a situation where mutations are almost unlikely to occur).

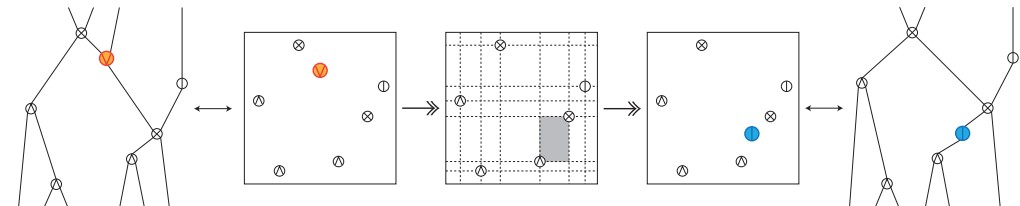

Figure 13: Illustration of simplest inference method for permutree process as marked point process. The current leveled permutree in Markov chain Monte Carlo inference corresponds to a certain state of the marked point process (**left**). One marked point (slightly enlarged and colored red) is chosen to be a candidate for updating. The region to be updated is quantized (**center**) to generate a new candidate marked point (slightly enlarged and colored blue) from the conditional posterior probability (or some proposal distribution). Finally, the generated new marked points are updated or not by the Metropolis-Hastings scheme, which is the next state of the Markov chain (**right**).

# D   Remaining challenges

The main difficulty in applying the permutree process to data modeling is how to handle its unlimited finite or infinite model complexity (i.e., number of vertices). Roughly speaking, it is not possible in principle to naively implement a model with infinite parameters on current computers. This is a

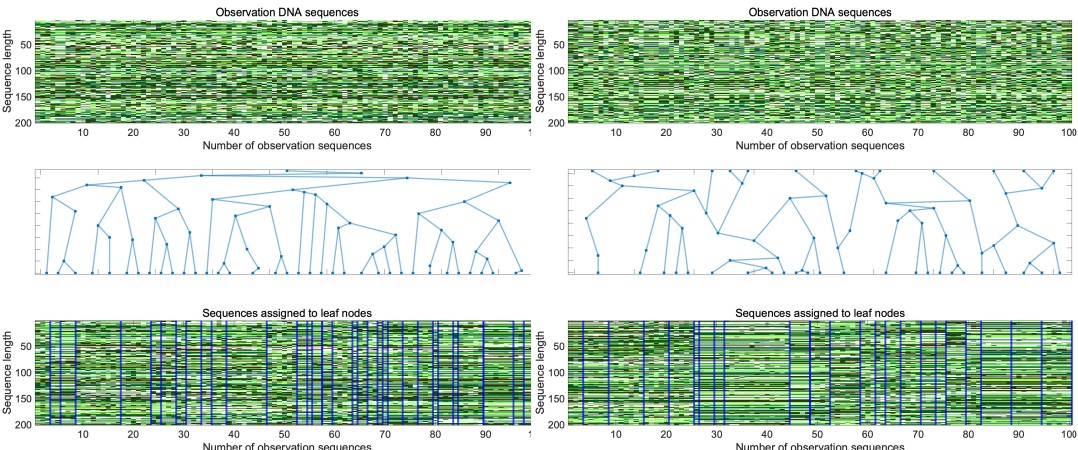

Figure 14: Binary tree (**left**) and Cambrian tree (**right**) attributed from permutree by restricting decoration weights.

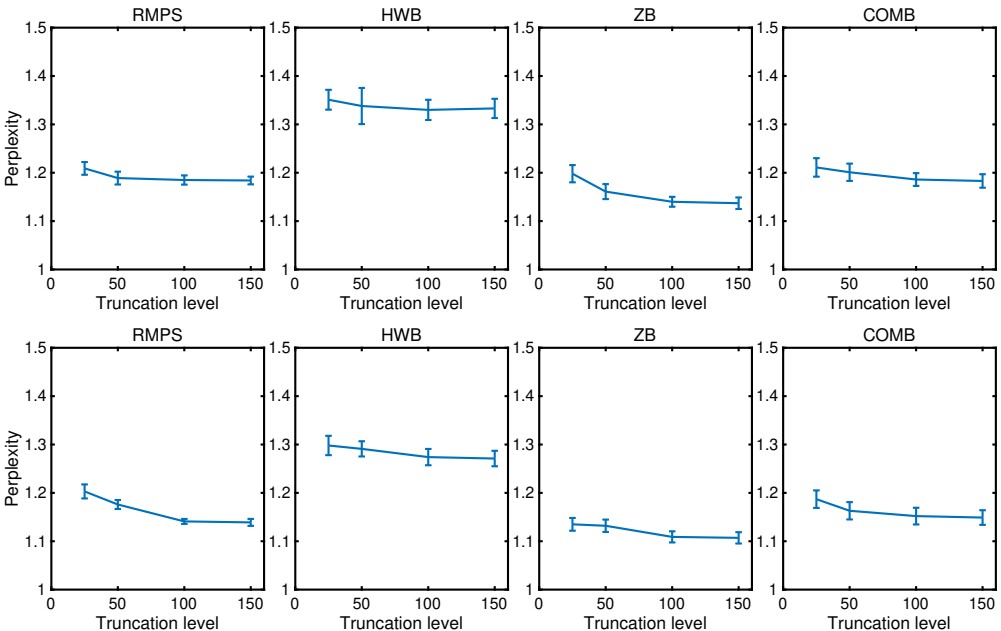

Figure 15: Effect of prediction performance on finite truncation level $K = 25, 50, 100$ and $150$ of marked stick-breaking process.

central topic in the BNP field, and we have historically had two policies. One is to represent models of infinite complexity such that finite truncation works reasonably well. This corresponds just to the representation methods for the stick-breaking process [89] in Dirichlet process infinite mixture models [79] and the beta-Bernoulli process in infinite factor models [95, 93]. The other method is a model representation in which, in conjunction with the finite amount of observed data, the model activates only as many of the potentially infinite number of parameters as necessary. This corresponds to the Chinese restaurant process [74, 76] in the mixure models or the Indian buffet process [29] in the factor models. While Section 4 focuses on the former policy, this section will explore the latter.

## D.1 Preliminary: ordered Chinese restaurant process

We begin our discussion with a representation using oCRP for binary trees, a special case of permutrees. Let $\theta > 0$ be the concentration parameters and $\alpha > 0$ the discount parameter. We will now take a so-called *spinal decomposition* (Figure 16 left) of the binary tree. In the metaphor of CRP,

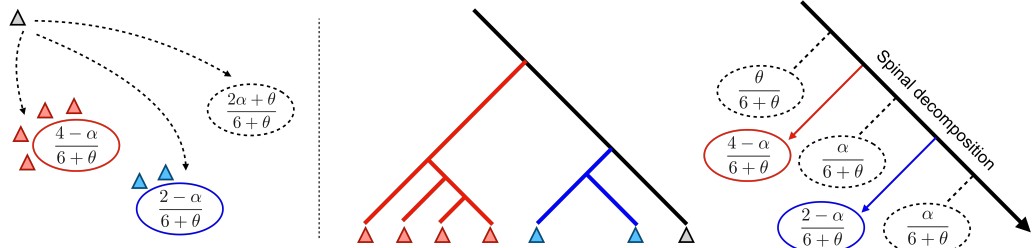

Figure 16: Standard Chinese restaurant process (left) for random partition and "ordered" Chinese restaurant process for random binary tree.

the customer can be viewed as as seeking a dish at the terminal node of the binary tree. The new customer can then either proceed to the existing subtree or create a new branch on one of the edges, according to the proportions shown in Figure 16 right. This can be viewed as CRP with random ordering of the CRP tables.

**Ordered Chinese restaurant process** (oCRP) [77, 85] - This stochastic process is a generative prbabilistic model that constructs a random binary tree by means of a recursive structure as follows. Let $\alpha$ and $\theta$ be the *discount* parameter and the *concentration* parameter, respectively.

- The first customer goes straight from the root to form one terminal node.

- The second customer forms a split between the root and the terminal node where the first customer is located. At this stage, the advanced subtree of the second customer is assigned a weight of $1 - \alpha$ and each edge of the split spinal cord is assigned a weight of $\alpha$ and $\theta$, respectively.

- The third and subsequent guests determine their own destination according to the proportion of weights assigned to the subtree and each edge on the spinal cord. If it chooses an edge on the spinal cord, it creates a new branch there to become a subtree and assigns weight $\alpha$ to the newly created edge on the spinal cord. If it moves on to an existing subtree, it recursively determines its own destination according to the nested oCRPs on that subtree and adds 1 to the weight of the subtree.

One will notice that this stochastic process is very similar to the standard Chinese restaurant process (CRP). If each subtree is considered a table, the probability that a new customer will sit at an existing table is proportional to the weight of the number of customers already sitting at that table minus the discount parameter $\alpha$. It will be seen that this is the same situation as in the standard CRP corresponding to the well-known Pitman-Yor process [76]. However, it differs from the standard CRP in that when a new table is seated, that table is determined by reference to the order of the existing tables. For this reason, this stochastic process is called an "ordered" CRP.

One of the most important properties of oCRPs is *exchangeablity*:

**Theorem D.1** (Proposition 1 (a) [77]). *A random binary tree generated by the nested oCRPs with the discount parameter $\alpha$ and the concentration parameter $\theta$ has exchangeable leaf labels for all $n \neq 1$ if and only if $\alpha = \theta = 1/2$.*

## D.2 Our attempt: Chinese restaurant street

**Strategy sketch and advance notice** - Recalling the requirements of (C1) and (C2) for the definition of permutrees (Section 2), we could introduce the following metaphor of CRP (See also Figure 17):

- Each *customer* is looking for a *dish* in a *Chinese restaurant street* and prefers a street that is popular with other customers, but is also willing to explore new streets on a whim. The development of the streets, with one customer after another searching for a dish, represents the permutree evolution.

- The Chinese restaurant street has a recursive structure. The *boulevard* (main street) has *side streets*, and each side street becomes the next boulevard, with its own next side streets recursively. This recursive structure would be reminiscent of an existing oCRP or nested CRP that recursively calls smaller CRPs in the overall process. When a side street becomes a cross street, it represents a vertex with $\otimes$. If the side street extends only one way, it corresponds to a vertex with $\oslash$ or $\obslash$.

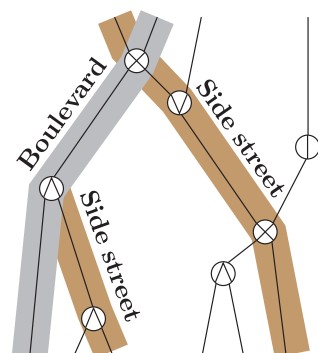

Figure 17: Overview of Chinese restaurant street

**Chinese restaurant street** (CRS) - CRS is given by the recursive structure of the streets, consisting of boulevards and side streets. Let $\theta_1 > 0$ and $\theta_2 \geq 0$ be the *concentration* parameters and $\alpha > 0$ the discount parameter. Figure 18 illustrates the situation where a new boulevard is a small CRS in a large CRS with the recursive structure. A CRS at a certain level consists of a boulevard and side streets, where each side of the boulevard is weighted by the concentration parameters $\theta_1, \theta_2$ and the discount parameter $\alpha$, and each side street is assigned a weight equal to the number of customers who proceeded to it minus the discount parameter $\alpha$. When the next customer enters this boulevard, the next destination is determined according to the ratio of those weights. It would have been a wishful idea if this vanilla CRS could be used as a permutree model, but unfortunately, it does not satisfy the requirements (C1) and (C2) as it is.

**Properties** - (1) The most important feature of CRS is that it is an extension of the existing oCRP. Specifically, in the case of $\theta_2 = 0$ (i.e., a situation where both boulevards and side streets grow only to one side), CRS is equivalent to oCRP for random binary trees. (2) Another important property is *exchangeability* (i.e., invariance of probabilities with respect to the order in which customers enter the process), which is often the case with variants of CRPs. For our CRS, we can show that it is *exchangeable* in the case of $\alpha = \theta_1 + \theta_2 = 1/2$, inheriting the result of exchangeability [77, Proposition 1] of oCRP. This property would be helpful in Bayesian inference.

**Theorem D.2.** *A random tree generated CRS with the discount parameter $\alpha$ and the two concentration parameters $\theta_1$ and $\theta_2$ (described in Section 3.2 of the main text) has exchangeable leaf labels for all $n \neq 1$ if and only if $\alpha = \theta_1 + \theta_2 = 1/2$.*

*Proof.* This can be verified by inheriting the exchangeability of oCRPs described in Theorem D.1. We shall consider each subtree in oCRP as a table and assign natural numbers of labels to the tables, starting from 1 according to the order in which the tables were generated. By reducing the resolution of the leaf labels in Theorem D.1 to table labels, the ordered tables generated by oCRP are also exchangeable. Then, for the random tree generated by CRS, if we consider each subtree as a table and set $\theta_1 + \theta_2 = \theta$, this is also attributed to the random ordered tables of oCRP with the discount parameter $\alpha$ and the concentration parameter $\theta$. Thus, by repeatedly applying the fact that the table labels of oCRP are exchangeable only when $\alpha = \theta = \theta_1 + \theta_2 = 1/2$, until each table eventually becomes a leaf node, we can confirm that the CRS has exchangeable leaf labels. $\square$

# E  Validity of transformation from marked points to permutree

This section verifies that the marked points $\{(l_i, m_i) : i = 1, 2, \ldots n\}$ generated from the marked point process are correctly transformed into permutrees by the algorithm in Figure 9 (Algorithm 1 of the main text). This can be verified by the following procedure, which is similar to the method in the proof of Proposition 8 in [75].

 (i) There is exactly one strand in each section separated along the auxiliary line (the red dotted line in Figure 9). This can be shown by mathematical induction on the number of nodes in the permutree.

 (ii) The graph created by the algorithm in Figure 9 has no cycles. This can be shown by using the proof by contradiction. If the graph had cycles, it would cross the red dotted line. However, by construction, the graph never crosses the red dotted line.

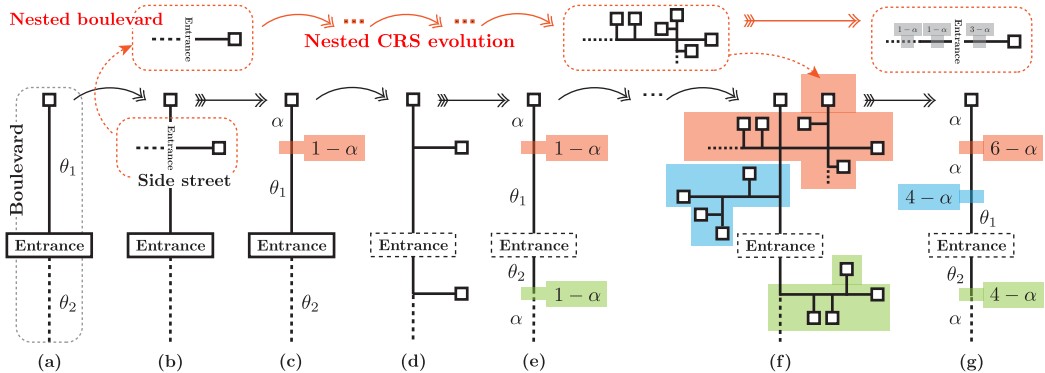

Figure 18: **Generative model of Chinese restaurant street** - **(a)**: Suppose a customer encounters a new boulevard. The boulevard has a *forward* road and a *backward* road on both sides of it with an entrance in between. When the new boulevard is opened up, which direction is the forward or backward road is determined with probability $1/2$. The figure shows the case where the forward road is up. Given two *concentration* parameters $\theta_1 > 0$ and $\theta_2 \geq 0$, the first customer to enter through the entrance chooses the forward road with probability $\theta_1/(\theta_1 + \theta_2)$, otherwise the backward road, and receives the dish being served at the end. We assign weights $\theta_1$ and $\theta_2$ to the forward and back roads, respectively. At this stage, the one that the customers did not choose between the forward and backward roads (the lower backward road, represented by the dashed line in the figure) has not yet been activated. Until both the forward and backward roads are activated, the entrance itself serves as another endpoint of this boulevard. **(b)**: The next customer coming to this boulevard, entering through the entrance to this boulevard, will proceed to the side street on that side in the proportion according to the weights assigned to each side. This side street itself corresponds to the boulevard in the next smaller CRS in the recursive structure. That is, the forward direction of this side street (i.e., whether it extends to the left or right first in the figure) is determined by this customer with probability $1/2$. The concept of whether the forward or backward road is chosen first on each side street determines whether the vertex corresponding to this side street in the permutree structure extends initially to the parent side or to the child side. **(c)**: Given a discount parameter $\alpha \geq 0$, this side street is assigned a weight of $1 - \alpha$, which is the number of customers who have taken the side street minus the discount parameter $\alpha$. The edge divided by the side street is assigned a weight of the discount parameter $\alpha$. **(d)-(e)**: When the next customer decides where to go based on the ratio of weights assigned to edges and side streets, she/he may choose an inactive road (dashed line in (c)). In that case, after the new side street is added (which also releases the entrance termination facility), the backstreet beyond it (dashed line in (d)) will continue to remain inactive. **(f)-(g)**: The above procedures are repeated sequentially and recursively.

(iii) From (i) and (ii), the graph generated by the algorithm in Figure 9 is a tree, and furthermore, since the red dotted line separates the left and right ancestor and descendant sub-trees, this is a leveled permutree.

