# OpenReview forum: "Permutree Process"
_NeurIPS.cc/2024/Conference — Submitted to NeurIPS 2024_

### Official Review · Reviewer_UGqF · 2024-07-04

**Soundness:** 2
**Presentation:** 3
**Contribution:** 3
**Rating:** 6
**Confidence:** 3

**Summary:**

This article adapts the recently-developed combinatoric concept of the “Pemutree” into a machine learning context, making links with existing methods in Bayesian nonparametrics, and providing a pathway for how to make the abstract mathematical concept relevant to data-driven approaches and inference. The theory is explained, and an example application in phylogenetic analysis is performed.

**Strengths:**

This is an ambitious paper! Making links with recent combinatorial research and machine learning is a good thing to do! It has a large vision of building one grand unifying theory of discrete Bayesian nonparametrics, which (it is claimed) can be done with the framework presented. The review of discrete BNP at the start of the article is thorough.

The potentially difficult subject matter of the abstract mathematical objects is explained fairly clearly through the use of figures and well-chosen notation. The theoretical aspects are explained thoroughly, and the application pursued emerges naturally from the framework developed earlier in the article: the coalescent analysis and the Mondrian process are good results to have. The scientific writing is of a fairly high standard, with only a couple of surprising vocabulary choices.

**Weaknesses:**

The subsequent developments of the theory and applications don’t quite live up to the grand vision set out earlier in the article. The authors struggle to represent the most widely-used discrete BNP object of the Dirichlet Process in this supposedly all-encompassing framework: perhaps this is doable in the future and represents work yet to be done, but the initial claims about the generality of permutree processes made in the article are not fully followed through on.

The article also (necessarily) spends a lot of time introducing the theoretical framework, quite heavily at the expense of presenting the data applications properly later in the manuscript. Squeezing all of the experimental results into “Demonstration” in half a page is really too brief to be very convincing, although there is a lot of interesting material in Appendix C that would ideally be in the main text. Many of the potentially thorny issues concerning inference and computation are therefore overlooked.

Some of the figures could be better designed: I found the visual interpretation of permutrees key to developing some understanding them, so making Figure 1 bigger and more prominent would be a help (I think figure 1 is more crucial than Figures 2/3/4 in this respect). The representation of the data in greenscale Figures 8 and 14 is very confusing: I don’t think I really learned anything from that representation.

Some of the language choices are a bit strange: line 69: “we dare to pay particular and explicit attention here”, line 869: “Roughly speaking, it is not possible in principle to naively implement a model with infinite parameters on current computers”,

Line 250: “as an overall trend…” The analysis of the experimental results is not rigorous enough. You have real values and uncertainties for the perplexity. Do some tests or similar to establish more clearly the differences in performances rather than painting broad brushstrokes.

Line 80: some of the symbols used have already established meanings in a machine learning context, i.e. \otimes meaning kronecker product. Maybe make clear that this is a new notation that overrides any previous perceptions.

**Questions:**

Line 219: “Data path modelling” the introduction of the two-table Chinese restaurant process is instructive, but might reveal the limitations of the model framework: are the only methods that can be captured by a permutree some variation on iterative applying of CRP-style binary splits and merges? Do you really think that this is a large enough “basic vocabulary” of operations to cover the entire world of discrete BNP methods out there? Is that a provable statement?
Line 336: epsilon = 0.01: where does that come from?

Appendix C.3: How would you describing performing the inference for this model? Did it need much fine tuning? Did it need lots of mixing time? Did it need a lot of computational resources? What about the comparison methods? Is this this “fancy but expensive” model? No shame if so, but it’s good to know this.

Line 887: “ordered Chinese Restaurant Process” - what are the material, practical implications of adding the ordering of tables on top of the standard chinese restaurant process? Can this model still answer the same type of statistical problems that a standard CRP does? Do the  categories/clusters observed in the data need to have some sort of ordering for this to be valid? Can the permutree represent a totally standard CRP without this ordering? If the permutree cannot represent the most widely used discrete BNP model then the claims towards the start of the paper about this being the grand unified framework need toning down.

Line 912: “Chinese Restaurant Street” - i worry that between the indian buffets and the Chinese Restaurant Franchises then the underlying metaphor is being overextended in this community. Why exactly have you proposed this model formulation? Do you consider it to be a useful new potential BNP model that has been underexplored so far? Is this the closest thing you can derive within this framework to a standard chinese restaurant process? This is unclear to me.

Random non-crucial question from an interested non-combinatorist: Can you define permutrees in a >2d space? This will surely change the topologically allowable relationships between nodes.

**Limitations:**

The practical models that are (presently) successfully captured in this framework are not the most widely used BNP models out there. The whole permutree framework seems best suited to the coalescent-type models pursued, and the other BNP models that have been successfully described, such as Mondrian processes, are interesting but not in widespread use.

---

> ### Author Rebuttal · Authors · 2024-08-04
>
> Thank you for your insightful and constructive comments. (To be more responsive to your important feedback, we would like to make additional remarks in 'Official Comment'.)
>
> ---
> **[Q1] Generality of our framework.**
>
> > “Data path modeling” (...) but might reveal the limitations of the model framework (...)
>
> This is an incredibly important issue that the BNP community has been facing for many years and is exactly what we wanted to discuss in this paper. Before discussing it in detail, we would like to start with a brief answer. Our argument is twofold:
>
> - *New insight*: Thanks to permutree, we have a unified modeling guideline for combinatorial stochastic processes, independent of the combinatorial structure of the target (permutation, tree, partition, binary sequence).
>
> - *Remaining challenge*: On the other hand, the projectivity and exchangeability (i.e., infinite exchangeability) required by the most BNP models must be achieved by us, the model designers, by cleverly using other mechanisms, such as the stick-breaking metaphor or the Chinese restaurant metaphor.
>
> **Background: difficulties in constructing new stochastic processes -** In Bayesian nonparametric community, the successful construction and design of essentially new combinatorial stochastic processes is often a brilliant and very important achievement, and has had a significant impact on subsequent research. However, creating essentially new combinatorial stochastic processes is a very difficult task. In fact, following achievements such as Dirichlet processes in classical statistics, the machine learning community has produced several essentially new stochastic processes, such as Indian buffet processes and Mondrian processes, which may have been largely due to a spark of genius by individual authors. More specifically, these successes have been made possible by a flash of genius in solving two challenges simultaneously:
>
> - (1) What generative algorithm can be used to represent a particular combinatorial object? (For example, in the Mondrian process, an algorithm that sequentially adds cuts to the block.)
>
> - (2) What probabilities can be assigned to that generative algorithm to make the model projective and exchangeable? (For example, in the Chinese restaurant process (CRP), the probability of table allocation should be a proportion of the number of customers sitting at the table up to that point.)
>
> **Benefits provided by permutrees -** The benefit of the permutree can be seen as giving a unified outlook to the former of these two challenges. More specifically, they give us the fact that various combinatorial structures (permutations, trees, partitions, binary sequences) can be represented as 'decorated permutations'.
> Thus, when modeling these combinatorial objects, we only need to introduce a generative algorithm with two functions: a mechanism that represents permutations and a mechanism that each has a decoration.
>
> **Remaining issues -** We will be able to use 'decorated permutations' as the basis for generative algorithms thanks to permutrees, but on the other hand, what probabilities can be assigned to them to have infinite exchangeability needs to be prepared separately again by the model designer. The permutree does not provide any new clues in this respect.
>
> The paper actually shows, as Section 4 shows, that the strategy of integrating the 'decorated permutation that is the permutree' with the stick-breaking and Chinese restaurant metaphor, a mechanism that automatically brings infinite exchangeability, is a promising one. Indeed, we believe that this strategy has some generality. By making the decoration possessed by the permutree a special case, it can be attributed to the standard CRPs, uniform permutations, Mondrian processes, etc. On the other hand, this strategy alone does not cover, for example, the Indian restaurant process. In this respect, it remains an open question as to what kind of mechanism can be used to provide infinite exchangeability.
>
> In light of the above, your concern about being overly assertive is valid: we would like to make a clear separation between the ambitions we are aiming for and what we are achieving at the moment by using the additional page given in the revised manuscript, if the paper is accepted.
>
> ---
> **[Q2] Relationship to conventional CRP and ordered CRP.**
>
> > “ordered CRP” (...) Can the permutree represent a totally standard CRP without this ordering? (...)
>
> The permutree process can be seen as a unified and general model that includes CRP and the ordered CRP (oCRP) as special cases. More specifically, we can attribute the permutree process (marked point process) to CRP and oCRP by restricting it so that only certain marks can appear in the permutree process. The table below illustrates the fact that, thanks to permutrees (= decorated permutations), these models can be represented in a unified way.
>
> | Model | Target combinatorial object | Exchangeability  | Decorations  |
> | ---- | ---- | ---- | ---- |
> | CRP | Partition | OK. | Restriction to (x) only |
> | Ordered CRP | Binary tree | OK. | Restriction to (v) only |
> | **Permutree process** | Permutree | OK. | **No restriction** |
>
> Additionally, the permutree we use also provides another interesting insight into the extension from CRP to oCRP, the traditional developments. Conventionally, oCRP has shown that by introducing a new table order = permutation for CRP, the representable objects can be extended from partitions to binary trees. This is very natural when one recalls that permutrees are exactly *decorated permutations*. In the context of this extension stream, our paper can be positioned within the following three stages of development.
>
> - For standard CRP, it can represent partitions.
>
> - For oCRP, the further introduction of **order** allows binary trees to be represented.
>
> - For permutree process, the introduction of **order** and **decoration** allows various combinatorial objects to be represented in a unified way.

---

> ### Author Response · Authors · 2024-08-07
> **Supplement to author response.**
>
> ---
> **[Q3] Motivation for Chinese restaurant street.**
>
> >  “Chinese Restaurant Street” (...) Why exactly have you proposed this model formulation?
>
> The motivation for Chinese restaurant street is to derive an alternative representation for stochastic processes with an inherently infinite dimensional parameter space that would work with only a finite number of parameters for a finite number of observations. This can be summarized as follows in contrast to the marked stick-breaking described in Section 4 of the main text.
>
> | Model | Target combinatorial object | Parameter dimension |
> | ---- | ---- | ---- |
> | Marked stick-breaking process | Permutree | Infinite (even for finite observation) |
> | Chinese restaurant street | Permutree-'like' | Finite (for finite observation) |
>
> This relationship is a frequently discussed contrast in Bayesian nonparametric methods, where two model representation methods have often been explored, for example in infinite mixture models for partitions and infinite factor models for binary sequences, as follows.
>
> | Model | Target combinatorial object | Parameter dimension |
> | ---- | ---- | ---- |
> | Stick-breaking process | Partition | Infinite |
> | Chinese restaurant process | Partition | Finite |
> | Beta-Bernoulli process | Binary sequence | Infinite |
> | Indian buffet process | Binary sequence | Finite |
>
> The main advantage of the Chinese restaurant street, the standard Chinese restaurant processes and the Indian buffet processes is that they do not require finite approximations in inference. In contrast, the marked stick-breaking process, the standard stick-breaking processes and the beta-Bernoulli process have infinite parameters with probability $1$, regardless of the size of the input observation data, so their parameter inference generally requires finite approximations as described in Section 4, and intricate mechanisms such as slice sampling.
>
> Finally, we would like to note that the current Chinese restaurant street is not a perfect model, as discussed in the appendix. The sample that this stochastic process could generate certainly has a permutree-like structure, but it does not fulfill the exact definition. More specifically, of the two requirements mentioned in Section 2 - (C1) that each interior point has one or two parents and (C2) that each interior point can be horizontally aligned - the latter is not straightforwardly satisfied. However, we believe that this modeling strategy is worth discussing in the appendix to this paper, as it follows a natural extension of CRP -> oCRP -> Chinese restaurant street.
>
> ---
> **[Q4] Practical computational cost.**
>
> Roughly speaking, our model is only a minor modification (adding a new 'decoration mechanism') of the standard stick-breaking process for the Dirichlet process infinite mixture model, so the substantial increase in computational complexity is not cause for concern. We add a diagram of `MCMC iterations $\times$ perplexity’ and ‘wall-clock $\times$ x perplexity’. As there was not enough space in the one-page PDF of the global response, we would like to update it directly in the revised version.
>
> ---
> **[Q5] Permutrees on high-dimensional space.**
>
> > Can you define permutrees in a >2d space? This will surely change the topologically allowable relationships between nodes.
>
> This is a very interesting topic. We don't have a direct answer to your question, but we may be able to offer some relevant and interesting insights. We are interested in what is the continuous analogy for a discrete structure of the permutree. In the machine learning community, the hyperbolic space is often mentioned as a continuous analogue to the binary tree (e.g., [Nickel&Kiela, NeurIPS2017]), which is a special case of the permutree:
>
> | Representation | Target combinatorial object | Space |
> | ---- | ---- | ---- |
> | Discrete | Binary tree | Ultrametric space |
> | Continuous | Continuous analogy of binary tree (continuous hierarchy) | Hyperbolic space |
> | Discrete | Permutree | Ultrametric / Euclidean |
> | Continuous | Continuous analogy of permutree | (Future work) |
>
> We expect that research could be developed in this direction in the near future. Very exciting question, thank you very much.
>
> - [Nickel&Kiela, NeurIPS2017] Poincare embeddings for learning hierarchical representations, NeurIPS2017.
>
> ---
> Finally, thanks again for your valuable comments.

---

> > ### Comment · Reviewer_UGqF · 2024-08-12
> >
> > This was illuminating, thank you.
> >
> > I think that clarification concerning how different existing BNP models fit within the permutree framework is important to really make the contribution of this article clear.

---

> > > ### Author Response · Authors · 2024-08-12
> > >
> > > We would like to thank you again for your kind comments and constructive feedbacks. As you mention, it is very important to clarify the existing BNP models within our framework. Your enlightening points have made this point much clearer. We really appreciate that we have been able to discuss this point in this valuable forum.
> > >
> > > With kindest regards,
> > >
> > > The Authors

---

### Official Review · Reviewer_mmW8 · 2024-07-05

**Soundness:** 4
**Presentation:** 3
**Contribution:** 3
**Rating:** 7
**Confidence:** 3

**Summary:**

After giving an introduction to permutrees, a stochastic process on them is constructed by sampling the nodes according to an intensity function on the 2d unit interval and uniformly assigning the marks. It is shown how to add the edges to meet the requirements for the object being a labeled permutree. Paths from terminal nodes to other terminal nodes in the permutree can be used to represent sequential data. Finally, the model is used for an inference task involving DNA sequences.

**Strengths:**

* Sections 1-4 are very concise but nonetheless clear and easy to follow. This paper does a great job explaining the complex concepts of permutrees and permutree processes.
* Figure 1-3 are very helpful
* the concept generalizes popular processes used in ML, e.g., the Mondrian processes

**Weaknesses:**

* As a reader unfamiliar with phylogenetic analysis, I did not immediately understand what the task in this setting is (and I am still unsure if I fully got it): Do you have a set of DNA sequences where some of the letters are masked, and you want to predict the masked letters?
* Besides not really understanding what the goal of this task is, I think Section 5 does not provide enough explanation of how this goal then is achieved, i.e., the length of the paper/ the level of detail in Section 5 is a problem. It's ok to defer details to the appendix as long as one can still follow the main section without them, but I struggle with that. Example given: I find it crucial to know the likelihood function when it comes to a Bayesian inference task, which is not mentioned in the main section.

**Questions:**

* What is the task in the phylogenetic analysis application?
* What would be other tasks where inference on permutrees is relevant in machine learning?

**Limitations:**

yes, there is a dedicated paragraph on limitations and I think it captures the limitations of the suggested concept appropriately.

---

> ### Author Rebuttal · Authors · 2024-08-04
>
> We are grateful for your important comments and suggestions.
>
> ---
> **[Q1] Task and goal of phylogenetic analysis.**
>
> > What is the task in the phylogenetic analysis application? Do you have a set of DNA sequences where some of the letters are masked, and you want to predict the masked letters?
>
> Your understanding is correct: The evaluation is based on the predictive performance of missing data. On the other hand, we take seriously the point that it took you some effort to understand it. We would therefore add the following explanation to further improve the clarity of our paper:
>
> Phylogenetic tree analysis can be seen as having the following two-stage goals:
>
> 1. *Informatics perspective*: to create better phylogenetic tree models and better inference algorithms.
> 2. *Scientific perspective*: to make scientific discoveries using better phylogenetic tree analysis methods.
>
> It is no exaggeration to say that the original ultimate goal of phylogenetic tree analysis is to discover the latter scientific findings. For example, in cases such as the recent pandemic (e.g., SARS-COV-2), phylogenetic tree analysis may reveal the pathways and causes of its spread. Therefore, as well as increasing the expressive power of models and improving the predictive performance of inference algorithms and the value of the objective function, it should also be emphasised that these are important applications that may lead to the discovery of scientific knowledge.
>
> ---
> **[Q2] Other applications of permutree process.**
>
> >  What would be other tasks where inference on permutrees is relevant in machine learning?
>
> We are delighted to be able to answer this important question. From an ambitious perspective, thanks to its generality, the permutree process is expected to provide new insights into machine learning applications where combinatorial models such as partitions, permutations, binary trees and binary sequences have traditionally been utilized. However, as stated in line 356 in the discussion of Section 6, we also recognize that this is not straightforward, since the permutree is only a 'prior model' and careful design of the 'likelihood model' is required for success in real applications. Here we would like to list some potential possibilities and some budding attempts.
>
> - (1) **Low-dimensional embedding** (like PCA, t-sne, UMAP, and GPLVM).
>
> An example of the highest generality is the potential application to low-dimensional embedding methods as an extension of the Gaussian process latent variable model. In general, the latent variable structure of the data requires the foresight knowledge of the model designer (e.g., a mixture model for partitions, or a tree structure model for differentiation and branching hierarchy). The permutree process prior model may play a role in data-driven inference of such latent variable structures. More specifically, the method assumes the permutree process as a prior model for the data as latent variables and a Gaussian process as a likelihood model from latent variables into observation data.
>
> - (2) **Density estimation** (like Dirichlet process mixture, Dirichlet diffusion tree, and Polya tree).
>
> Another highly generalized task is the application to density estimation tasks. Indeed, various Bayesian non-parametric models have traditionally been applied to this task, including Dirichlet process mixture models, Dirichlet diffusion trees and Polya trees. With the similar motivation as in the above low-dimensional embedding task, we can think of ways to model the density function in a way that assumes permutree processes in the latent variable structure of the observable density function.
>
> - (3) **Dictionary learning** (like beta-Bernoulli process and Indian buffet process).
>
> One of the more concrete applications is dictionary learning. We consider, for example, the task of learning a dictionary of images. Here, an image can be regarded as a collection of vectors by converting a small patch (e.g. 8x8 pixels) into a vector (64-dimensional vector). The task of learning a dictionary from this data by means of factor analysis is one of the standard tasks in machine learning. In the Bayesian nonparametric field, infinite factor models based on the Indian buffet process or the beta-Bernoulli process are often used.
>
> Our focus here is to assume a permutree structure within this collection of factors. Indeed, it has been reported in the past that the introduction of cluster or hierarchical structures within factors, e.g. by exploiting spatial correlations within an image, can be effective. We may therefore introduce a mechanism for data-driven inference of such dependencies between factors by means of the permutree process prior model. This is illustrated in 'one-page PDF in global responses'.
>
> ---
> **[Q3] Advice on improving clarity.**
>
> > Besides not really understanding what the goal of this task is, I think Section 5 does not provide enough explanation of how this goal then is achieved, i.e., the length of the paper/ the level of detail in Section 5 is a problem. It's ok to defer details to the appendix as long as one can still follow the main section without them, but I struggle with that.
>
> We will be able to meet your request by specifying more details of the application cases in this paper, thanks to the additional content page given in the camera-ready version, if this paper is accepted. We are grateful for constructive advice on how to raise the clarity of our paper.

---

> > ### Comment · Reviewer_mmW8 · 2024-08-12
> >
> > Thank you for the clarifications. I was a bit unsure about the relevance for the machine learning setting, but the answer to Q2 makes me more confident in that regard. I raise my score from 6 to 7.

---

> > > ### Author Response · Authors · 2024-08-12
> > >
> > > We would like to thank you again for your valuable time and thoughtful comments. We are particularly pleased to have this valuable opportunity to discuss with you the potential applications of machine learning.
> > >
> > > With kindest regards,
> > >
> > > The Authors

---

### Official Review · Reviewer_V6GE · 2024-07-08

**Soundness:** 3
**Presentation:** 1
**Contribution:** 3
**Rating:** 5
**Confidence:** 3

**Summary:**

The authors introduce a prior for Bayesian nonparameterics called the permutree. They apply it to model complex phylogenetic data with both coalescence and recombination, a setting that previous processes such as the Kingman could not model; the model seems to perform state-of-the-art phylogenetic inference. In principle the process could also be used to model other combinatorial objects such as permutations or trees however this is not demonstrated.

**Strengths:**

Modeling recombination is challenging and this model proposes a method to do so.

**Weaknesses:**

The writing is very challenging. There are multiple points where the writing is strange, for example the use of "dare" in "For technical reasons (discussed immediately below), we dare to pay particular and explicit attention here to the set V of the “interior vertices” (i.e., vertices of degree at least 2) other than the terminal nodes." ". The exposition is also very verbose and proposition is challenging to understand without reading the proof. Figures 3(c) and 1(b)-(d) are never mentioned in the text, the later is quite confusing since it seems to suggest that the permutree can model many combinatorial objects.

It seems that there is a disconnect between the description of the methods in sections 3, 4, and the experiment in section 5. See questions.

The ultimate goal of phylogenetics is to infer ancestry which is not identical to maximizing likelihood. To validate a bone fide phylogenetic inference method that handles recombination, one should show that inferred recombination events are realistic.

**Questions:**

In section 3 you state that you generate $l$ from a Poisson process then in section 4 you use a stick-breaking procedure. Which do you use in practice? In the genetics case, $l_k$ is stated to be drawn from a point process in eqn 7. if this is the case, is the data still modeled as part of an infinite exchangeable process? How could I query the posterior predictive for example?

In the genetics case, could you explain the role of the two table Chinese restaurant process? It seems that sequences are clustered by node and that the path does not affect their likelihood.

What is the form of the likelihood for recombination?

**Limitations:**

Discussed.

---

> ### Author Rebuttal · Authors · 2024-08-04
>
> Thank you so much for your important comments. (We would like to be more responsive to your important comments, so let us make additional remarks in the 'Official Comment' section.)
>
> ---
> **[Q0] More concise guidance.**
>
> > The writing is very challenging.
>
> Thank you for your helpful advice. By refining each lead sentence to more succinctly present the subject of each paragraph, we improve the revised version to be more readable for a diverse readership.
>
> ---
> **[Q1] Ultimate goal of phylogenetic tree analysis.**
>
> > The ultimate goal of phylogenetics is to infer ancestry which is not identical to maximizing likelihood.
>
> We agree with this opinion. Thank you for your very important remarks. Building on this advice, we would like to clarify the objectives of this paper in an applied perspective by adding the following explanations. More specifically, **we would like to clarify the original purpose and goal of phylogenetic tree analysis in two steps as follows and which part of it this paper is trying to focus on.**
>
> In general, phylogenetic tree analysis can be seen as having the following two-stage goals:
>
> 1. *Informatics perspective*: to create better phylogenetic tree models and better inference algorithms.
>
> 2. *Scientific perspective*: to make scientific discoveries using better phylogenetic tree analysis methods.
>
> As you point out, the original ultimate goal of phylogenetic tree analysis is to discover the latter scientific findings. For example, in cases such as the recent pandemic (e.g., SARS-COV-2), phylogenetic tree analysis may reveal the pathways and causes of its spread. From this perspective, increasing the expressive power of the model or improving the predictive performance of the inference algorithm or the value of the objective function are only milestones towards achieving the ultimate goal.
> As you point out, both of these aspects are significantly important in the development of science.
> At this time, we expect that many NeurIPS readers in the machine learning community are interested in this former perspective. Therefore, this paper also presents an evaluation of each model/algorithm using objective predictive performance on missing data. At the same time, however, it is important to explicitly state that this is aimed at the ultimate goal beyond this - the discovery of scientific knowledge. We are deeply grateful for the very enlightening advice.
>
> ---
> **[Q2] Form of the permutree process in practice.**
>
> > In section 3 you state that you generate $l$ from a Poisson process then in section 4 you use a stick-breaking procedure. Which do you use in practice?
>
> Thank you for your important question on improving the clarity. **In practice we use the marked stick-breaking process (in Section 4 of main text), which correspond to a special case** of the general concept of a permutree process, i.e., the marked point process (in Section 3 of main text).
>
> In response to this point of view, we would like to improve the presentation of our paper. In the version at the time of submission, we attempted to present modeling and inference methods in a more generalized situation, so that the reader can easily adjust the intensity function of the marked point process according to the individual task. On the other hand, this may be a weaker way of conveying the appeal of the marked stick-breaking process, which is the special case we most wish to recommend. We therefore intend to address the following in our revised paper.
>
> - *Clarity of motivation*: In Section 4, we add an explanation of the qualitative reasons for recommending the marked stick-breaking process in practice.  More specifically, we clarify that the stick-breaking process is a prior model that induces sparser analysis results and is expected to have the effect of suppressing the generation of useless nodes to some extent. (Needless to say, the acceptability of this qualitative intuition is a topic that has been deeply studied theoretically and is very difficult to handle, so we explain it carefully to the reader.)
>
> - *Clarification of modification procedure*: We specify which parts of the model/inference need minor modifications when adopting the marked stick-breaking process, in addition to the general marked point process formulation.
>
> - *Supplementary material*: We add the following items to the experimental results as supporting evidence that marked stick-breaking processes give better empirical results than marked point processes with uniform intensity, as shown in '*one-page PDF in global responses*'
>
> ---
> **[Q3] Infinite exchangeability (projectivity and exchangeability) of the general permutree process described in Section 3.**
>
> > if this is the case, is the data still modeled as part of an infinite exchangeable process? How could I query the posterior predictive for example?
>
> **Yes, most of the properties of the marked stick-breaking process (we described in Section 4) can also be inherited by the marked point process described in Section 3.** Therefore, the inference of predictive and posterior distributions is similarly tractable. More specifically, it can be summarized as follows.
>
> - The marked point processes, permutree processes, are projective. This follows immediately from the projective nature of Poisson processes.
>
> - If we build the model of the datapath described in Section 4 by a general marked point process instead of a marked stick-breaking process, it is still exchangeable. Therefore, in terms of datapaths, the model of a datapath has infinite exchangeability, since exchangeability induces projectivity as it is.
>
> One point to mention where the general marked point process differs from the special marked stick-breaking process is that when its intensity function is finitely restricted, the number of interior points of the random permutree as a prior distribution is finite with probability $1$ (an unlimited finite, but not infinite, situation).

---

> ### Author Response · Authors · 2024-08-07
> **Supplement to author response.**
>
> ---
> **[Q4] Role of two-table Chinese restaurant process.**
>
> > In the genetics case, could you explain the role of the two table Chinese restaurant process? It seems that sequences are clustered by node and that the path does not affect their likelihood.
>
> Thank you for your important question. We would like to address your concern as follows:
>
> The role of the two-table Chinese restaurant process (2tCRP) serves to bring together DNA sequences that follow similar lineages (evolution over time) within a collection of observed DNA sequences. The tables that our 2tCRP have can be regarded as checkpoints for events (such as coalescence and recombination) in the time direction (vertical in the diagram) in gene evolution. The table serves to group together DNA sequences that can be regarded as having behaved identically at the checkpoints by means of table assignments. Our 2tCRP allows a collection of observed DNA sequences to be stored as data paths on a permutree to define a prior distribution. The likelihood of these data paths is then evaluated by using the evolutionary models such as the Jukes-Cantor model and the generalized time reversible model.
>
> This modeling strategy can also be seen as an extension of the Dirichlet diffusion tree [Neal2003] for representing binary trees to a new model for representing permutrees. Dirichlet diffusion trees also represent data diffusions in a similar way to the table assignment probabilities we use when an event occurs. We would like to add an explanation of the motivation for the 2tCRP in terms of this extension of the Dirichlet diffusion tree.
>
> - [Neal2003] Radford M. Neal. Density modeling and clustering using Dirichlet diffusion trees. Bayesian Statistics, 7:619–629, 2003.
>
> ---
> **[Q5] Likelihood for recombination.**
>
> > What is the form of the likelihood for recombination?
>
> The recombination is just selecting a split position uniformly at random. That is, the likelihood model of recombination is the concatenation of two DNA subsequences at the mutation-free, non-informative uniform split position. More specifically, as a generative probabilistic model, given two sequences (e.g., ACGTTC and GCGTCA), the recombination can be viewed as a stochastic operation that
>
> - The split position ( **** , ** ) of the sequence ( ****** ) is generated uniformly at random.
> - The two sequences are converted into a sub-sequence ( ACGT** ) leaving only the left side and a sub-sequence ( ****CA ) leaving only the right side of this split position, which are then concatenated and recombined into a single sequence ( ACGTCA ).
>
> It should also be emphasized that the likelihood model other than this uniform random split is separately represented by the evolutionary models. The recombination is assumed to be instantaneous and not accompanied by mutation. This means that the likelihood model for genetic mutation is separately represented by the JC and GTR models.
>
> ---
> Finally, thanks again for your valuable comments.

---

> > ### Comment · Reviewer_V6GE · 2024-08-07
> > **response**
> >
> > Thank you for clarifying. Due to the challenging presentation and challenges in validating the genetics modeling case. I keep my score the same.

---

> > > ### Author Response · Authors · 2024-08-07
> > >
> > > We thank you again for your time and thoughtful comments. We are very appreciative of your comments, which have helped us to increase the clarity of our revised version even more.
> > >
> > > We remain available in case any additional query arises.
> > >
> > > With kindest regards,
> > >
> > > The Authors

---

### Official Review · Reviewer_7WLf · 2024-07-12

**Soundness:** 3
**Presentation:** 3
**Contribution:** 3
**Rating:** 6
**Confidence:** 2

**Summary:**

The authors describe the concept of permutrees, how to sample permutrees in a stochastic process, and how to model data with permutrees. They apply it to tracking DNA changes.

**Strengths:**

Interesting new model that unifies permutations, trees, partitions, and binary sequences

Strong mathematical foundation

Practical applications

well-written

**Weaknesses:**

the figures are small and hard to read when printed in gray-scale

**Questions:**

I do not know what a stick breaking process or Chinese restaurant process is?

>Fig.1 (l)

It is not obvious how the binary number is generated.
Could one say if the line from one node to the next goes up it yields 1 and if it goes down it yields 0?

>p7 253 numnber

number ?

>p7 261 "tiny real value"

how tiny is tiny ?

>p16 669 uniformly random random random variables Uj

is that more random than just a random variable?

**Limitations:**

yes

---

> ### Author Rebuttal · Authors · 2024-08-04
>
> We appreciate your helpful comments and recommendations.
>
> ---
> **[Q1] Improvements to color schemes and size in figures.**
>
> > the figures are small and hard to read when printed in gray-scale
>
> Thank you for your important advice. We will improve the color scheme we used in our diagrams so that the colors are identifiable even when displayed in grayscale. Also, if this paper is accepted, we would like to use the extra page given to the camera-ready version to display the important figures in a larger size.
>
> ---
> **[Q2] Supplementary information on the basic tools of Bayesian non-parametric methods for a diverse readership.**
>
> > I do not know what a stick breaking process or Chinese restaurant process is?
>
> Thank you for raising this important point. We want this paper to be read by a diverse audience, not just the BNP community, so we would like to add some background knowledge to the basic tools in the appendix.
>
> Both the stick-breaking process and the Chinese restaurant process have traditionally been used in the BNP community for more than 20 years as the most fundamental tools for representing the 'random partitions'. The properties of these two models can be summarized as follows:
>
> | Model | Target combinatorial object | Exchangeability | Projectivity | Parameter dimension |
> | ---- | ---- | ---- | ---- | ---- |
> | Stick-breaking process | Partition | OK. | OK. | Infinite |
> | Chinese restaurant process | Partition | OK. | OK. | Finite (for finite observation) |
>
> Projectivity and exchangeability are generally used as the two conditions required for BNP models. Roughly speaking, these can be seen as conditions that guarantee that the BNP model can be defined as a stochastic model on an infinite dimensional parameter space. Plainly, *exchangeability* refers to the requirement that the labels of the target objects to be represented (e.g., the index of the data) does not affect the probability of the model. *Projectivity* refers to the self-similarity of the model with respect to the size of the target objects.
>
> **Reason for our use of these tools.**
>
> In this paper, we aim to exploit the fact that the stick-breaking process and the Chinese restaurant process provide projectivity and exchangeability in random *partitions* and extend it to a more general case, random *permutrees*. Therefore, we make frequent use of these tools in the paper.
> | Model | Target combinatorial object | Exchangeability | Projectivity | Parameter dimension |
> | ---- | ---- | ---- | ---- | ---- |
> | Marked stick-breaking process | **Permutree** (that contains partition as a special case) | OK. | OK. | Infinite |
>
> Thank you for your important remarks. Thanks to your advice we can improve our exhibition methods for a diverse readership.
>
> ---
> **[Q3] Binary sequence as a special case of permutree.**
>
> > For Fig. 1(l), It is not obvious how the binary number is generated. Could one say if the line from one node to the next goes up it yields 1 and if it goes down it yields 0?
>
> Your understanding is perfectly correct. Just to be sure, we would like to reiterate Figure 1(l) in the main body of the paper here as well.
>
> We read the vertical indices of the permutree in Figure 1(l) (shown in blue) from left to right, yielding '462153'. We can obtain a binary sequence by comparing the size relationship between adjacent left-right pairs of this sequence from the beginning and outputting '0 if the left side is larger and 1 if the right side is larger'. More specifically,
>
> - 4<6: => Output: 1
>
> - 6>2: =>Output: 0
>
> - 2>1: => Output: 0
>
> - 1<5: => Output: 1
>
> - 5<3: => Output: 0
>
> As a result, Fig. 1 (l) can obtain the binary sequence "10010".
>
> ---
> **[Q4] Meaning situation in $\epsilon$**
>
> In practice, it makes sense to have a situation where $\epsilon<1/N$ for the size $N$ of the observation data. On the other hand, more precisely, the propositional assertion itself holds for any $0<\epsilon<1$. This is a point we wish to make intelligible.

---

### Author Rebuttal · Authors · 2024-08-06

---
**Thanks to all involved. -**
We are very grateful to all the Reviewers who spend their valuable time to read our papers and give us constructive and favorable comments and suggestions. We are also deeply grateful to the Area Chairs (ACs) and Program Chairs (PCs) who, through their professional management, have given us this valuable opportunity.

---
**Supplementary one-page PDF. -**
We refer to the more intuitive diagrams for supplementary purposes when responding to some of the questions of each reviewer. These are referred to in our responses as '*one-page PDF in global responses*' and we hope you will make use of them.

---
**Wish to address additional concerns as well. -**
We will be pleased to be able to address any new points of concern that are noticed during the 'discussion phase until 13 August' to resolve them. We would like to gratefully appreciate in advance the time that reviewers, ACs and PCs will continue to devote to our paper during the subsequent management.

---

### Decision · Program_Chairs · 2024-09-25

**Decision:**

Reject

**Comment:**

Summary of recommendation: Reviewers' scores for this paper were relatively high, but the actual content of their reviews was scant on strengths and reported a high degree of uncertainty and confusion about the paper's content. From the reviews, I was not convinced that the reviewers had been able to meaningfully evaluate the paper, partly due to the fact that much of the paper's technical content appears in the supplementary material. I read the paper myself and agree that the important ideas are very difficult to understand and evaluate from the main text, and believe the paper needs a major revision before it can be properly reviewed/evaluated at an ML conference.

This paper draws on a new concept in combinatorics called a permutree to construct a novel stochastic process---the permutree process---which can represent a range of "combinatorial structures such as permutations, binary trees, partitions, and binary sequences". The paper proposes this process as a Bayesian nonparametric prior which generalizes and unifies a large number of existing ones (e.g., CRP, IBP, Mondrian process, among many others). The reviewers were all impressed by the paper's sweeping generality, one reviewer saying: "an ambitious paper! [...] It has a large vision of building one grand unifying theory of discrete Bayesian nonparametrics".

Several aspects of the paper are very clear. Its grand vision of a unifying framework of BNP processes as permutree processes is clear. What exactly a permutree is, along with other prerequisite concepts in combinatorics, is explained clearly (reviewers appreciated this). What exactly the proposed permutree process is---i.e., a marked Poisson point process on $[0,1] \times [0,1]$ with marks taking values in the set {$\vee, \wedge, |, \times$}---is also clear.

Several important aspects of the paper are not clear. What data analysis actually looks like using the permutree process is very hard to grasp from the main text. For instance, the main text does not include any example of a full model described mathematically according to its generative process. The paper reports one case study, where it builds a model based on permutree processes for phylogenetic tree analysis. However, basic facts about the model (e.g., its likelihood) are not described sufficiently, and appear only in highly condensed prose. After reading the main text, I was unable to write down for myself what the actual model discussed in Section 5 was; reviewers made similar comments to this effect. (Many of these details do appear in the supplementary material, including a full generative process, and a detailed description of the model's rationale; I'll return to this later.)

Another important aspect of the paper that is not made clear by the main text is whether the proposed framework really does succeed at unifying BNP processes under a useful abstraction. Reviewers complained about a lack of any clear example showing how a well-known process (e.g., a CRP) is an instance of a permutree process. For instance, the same reviewer quoted above goes on to say: "The subsequent developments of the theory and applications don’t quite live up to the grand vision set out earlier in the article. The authors struggle to represent the most widely-used discrete BNP object of the Dirichlet Process in this supposedly all-encompassing framework [...]". (Again, many more details along these lines do appear in the supplementary material.)

Finally, taking as given that the proposed abstraction does indeed generalize most well-known BNP processes, the paper does not clearly demonstrate a benefit in doing so. I can imagine several potential benefits of an abstraction that unifies many BNP processes: one might use that abstraction to prove something that applies to many known models, one might use that abstraction to craft new BNP processes whose properties can be easily characterized, one might use that abstraction to build a general-purpose inference scheme, the list goes on. While the paper does build one model using the new abstraction, the details of that model are difficult to understand and evaluate, and the benefits of the abstraction are not clearly illustrated.

The lack of clarity that the reviewers commented on and that I have written about here is due mainly to the way the paper is structured, leaving critical details and necessary exposition in the supplementary material. There are  no guidelines at NeurIPS about how to structure a submission; however reviewers at NeurIPS (and most ML conferences) are not required to read the supplementary material, and often don't. In this case, I do not think the paper is understandable from the main text, and am not convinced that the reviewers were able to properly evaluate it. The paper reads like a journal article that was crammed into a conference paper. I think this paper needs to be rewritten so that its contributions can can be illustrated, demonstrated, and appreciated in the main text before it is ready for acceptance at a venue like NeurIPS.